# Recent Criterion on Stability Enhancement of Perovskite Solar Cells

Md Saif Hasan [1,†], Jahangir Alom [1,†], Md Asaduzzaman [1], Mohammad Boshir Ahmed [2], Md Delowar Hossain [3], ASM Saem [4], Jahangir Masud [5], Jivan Thakare [5,*] and Md Ashraf Hossain [5,*]

1   Department of Applied Chemistry and Chemical Engineering, University of Rajshahi, Rajshahi 6205, Bangladesh; s1710275117@ru.ac.bd (M.S.H.); s1710875124@ru.ac.bd (J.A.); s1810875101@ru.ac.bd (M.A.)
2   Department of Materials Science and Engineering, Gwangju Institute of Science and Technology, Gwangju 61005, Korea; mohammad.ahmed@gist.ac.kr
3   SUNCAT Center for Interface Science and Catalysis, Department of Chemical Engineering, Stanford University, Stanford, CA 94305, USA; dhossain@stanford.edu
4   Department of Chemistry, Temple University, Philadelphia, PA 19122, USA; tui79287@temple.edu
5   Energy and Environmental Research Center, University of North Dakota, Grand Forks, ND 58202, USA; jmasud@undeerc.org
*   Correspondence: jthakare@undeerc.org (J.T.); ahossain@undeerc.org (M.A.H.)
†   These authors contributed equally to this work.

**Abstract:** Perovskite solar cells (PSCs) have captured the attention of the global energy research community in recent years by showing an exponential augmentation in their performance and stability. The supremacy of the light-harvesting efficiency and wider band gap of perovskite sensitizers have led to these devices being compared with the most outstanding rival silicon-based solar cells. Nevertheless, there are some issues such as their poor lifetime stability, considerable *J–V* hysteresis, and the toxicity of the conventional constituent materials which restrict their prevalence in the marketplace. The poor stability of PSCs with regard to humidity, UV radiation, oxygen and heat especially limits their industrial application. This review focuses on the in-depth studies of different direct and indirect parameters of PSC device instability. The mechanism for device degradation for several parameters and the complementary materials showing promising results are systematically analyzed. The main objective of this work is to review the effectual strategies of enhancing the stability of PSCs. Several important factors such as material engineering, novel device structure design, hole-transporting materials (HTMs), electron-transporting materials (ETMs), electrode materials preparation, and encapsulation methods that need to be taken care of in order to improve the stability of PSCs are discussed extensively. Conclusively, this review discusses some opportunities for the commercialization of PSCs with high efficiency and stability.

**Keywords:** perovskite solar cells; stability; moisture; degradation; encapsulation

## 1. Introduction

Due to the rapid socio-economic development of the world, the global energy consumption has been gradually increasing. At present, fossil fuels mostly meet the demand for energy but the inadequate supplies of fossil fuels require research into sustainable and renewable energy. Solar energy is one of the most promising alternatives to fossil fuels due to its easy conversion of sunlight into electricity and low negative impact on the environment. The most common form of solar energy use is that of photovoltaic (PV) cells. The first photovoltaic (PV) solar cell was developed in 1954 [1]. From the beginning, researchers intended to design PV devices with high power conversion efficiency (PCE), large-scale fabrication potential, low cost, and environmentally friendly in nature for commercialization but remain to succeed in this goal to date.

Photovoltaic solar panels can be classified into three groups. The very first group was named first-generation silicon wafers solar cells. Though these are more ancient than the

rest of PV panels, its high power efficiency and stability have made it the most widely commercialized among PV devices. The following type is called second-generation solar cells, which include thin films solar cells and proved themselves to be more efficient than the first generation. The last group only includes a few under-researched and still emerging PV cells such as nanocrystal-based solar cells, polymer-based solar cells, dye-sensitized solar cells, perovskite-based solar cells, and concentrated solar cells [2].

Recently, perovskite solar cells (PSCs) have attracted significant interest in recent years in the field of optoelectronics because of their economically and environmentally beneficial as well as renewable characteristics, presenting an alternative to traditional solar cell technologies and an opportunity to address global challenges in energy generation and climate change [1]. Several classes of perovskite materials such as chalcogenide perovskite ($AMO_3$), halide perovskites ($AMX_3$), and hybrid perovskite have been discovered but among them, organic–inorganic halide perovskites have been successful so far in outshining other classes. They are a group of materials with a composition of $AMX_3$, where A is an organic cation methylammonium ion ($MA^+$), formamidinium ion ($FA^+$), and inorganic metal cation cesium ($Cs^+$). M is a divalent metal cation ($Pb^{2+}$, $Sn^{2+}$), and X refers to monovalent anions ($I^-$, $Br^-$, $Cl^-$ or $SCN^-$) (Figure 1) [2]. In the 1990s, perovskite materials were first investigated in the field of optoelectronics, and they showed strong exciton features and were eventually used in the field of LEDs, transistors, and solar cells [3,4]. Koijima was the first to report $CH_3NH_3PbI_3$ and $CH_3NH_3PbBr_3$ perovskite materials as liquid sensitizers in a dye-sensitized solar cells structure in 2009 [5]. However, these solar cells attained very low PCE (3.81~3.20%) and poor stability. However, since then, a convincing improvement in PCE of such devices has been noticed [6]. The abrupt development of PCE from 3% to 25.2% (28% in tandem architecture) within just the past 10 years has proved the PSCs to be a promising type of power source for the future, while other technologies took nearly 30 years to witness this milestone.

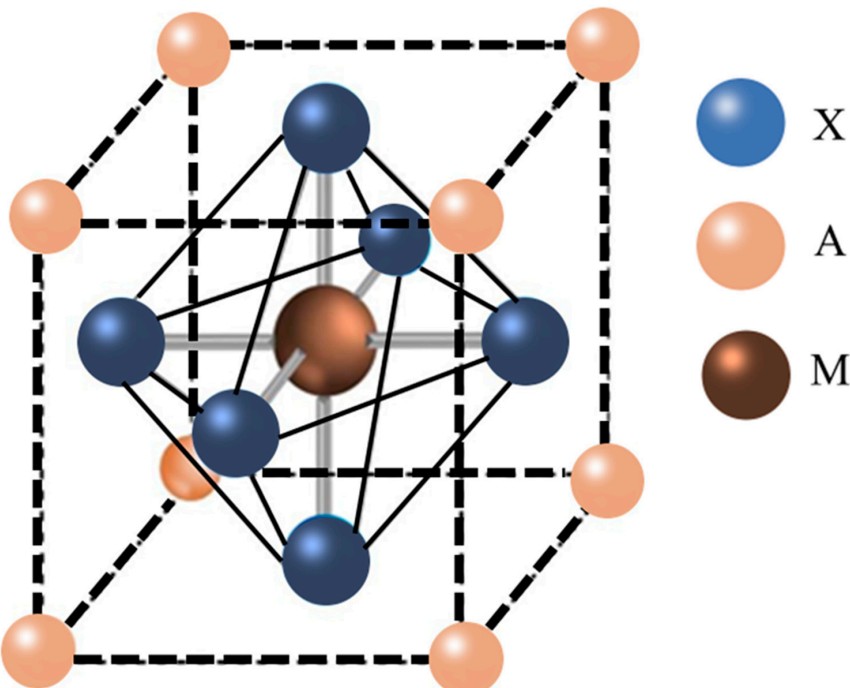

**Figure 1.** A typical perovskite structure.

Although the efficiency of these PVs has gradually increased day by day, the main obstacle to commercializing them in the market is their poor lifetime stability. It has always been predicted that a PV would exhibit a steady-state performance and stability for at least 25 years of its storage and continuous operation [6]. However, due to the various characteristic structural and chemical properties of the device components, perovskite PVs have

rarely demonstrated such stability in recent years. For instance, light-harvesting perovskite materials, which are significant components of a PSC device, are mostly hygroscopic. In the presence of moisture under operating conditions, they easily become hydrolyzed within a short time and cause the robust degradation of the PVs [7,8]. Additionally, another problem arises from the presence of UV light under real operating conditions. In addition to the desired transportation of the holes and electrons, some decomposed ions of the light-harvesting perovskite material also undesirably migrate from their enacting locations in the perovskite crystal structure when obtaining sufficient activation energy ($E_a$) for ion migration [9–11]. In fact, this is an obstinate phenomenon of the perovskite layer of a PSC device that considerably reduces its lifetime.

Furthermore, according to the state-of-the-art structure of PSCs (i.e., most typically, planar n–i–p or p–i–n heterojunction structure), the electron transport layer demands a good electron conducting material with enhanced electron mobility and well-matched energy band alignment with the perovskite layer that can efficiently extract electrons from the light-harvesting layer and frequently transport them to the electrodes [12]. In this regard, most researchers elected $TiO_2$ and contributed to attaining the most efficient PVs to date with conventional or simply modified perovskite light harvesters [13–16]. Nevertheless, as many researchers have elicited and confirmed, $TiO_2$ can act as an effective photocatalyst for organic compounds and cause the robust degradation of the perovskite layer of PSC devices under continuous light illumination [17,18]. Another promising candidate in this context, namely ZnO, demonstrated some significant thermal degradation under the processing and operating conditions of PV devices [19,20]. Furthermore, the considerable degradation of the perovskite layers of the ZnO–electron transport layer (ETL)-based PSC has also been reported by many researchers due to the instant chemical reactions between the ZnO and the respective perovskite layer [21,22].

In contrast, the hole transport layer (HTL), another essential part of PSCs, demands an excellent hole-conducting material with enhanced hole mobility and energy level compatibility with its surrounding layers. In this context, an organic–inorganic hybrid state-of-the-art material, e.g., 2,2′,7,7′-tetrakis-(*N*,*N*-di-p-methoxy-phenyl-amine)-9,9′-spiro-bifluorene (spiro-OMeTAD), has been proposed by most researchers to obtain an enhanced device performance with their fabricated PVs [23–25]. However, it was later proven that, under operating conditions, the formation of pinholes inside this material may be another significant factor obstructing the long-term steady-state performance of the devices [26,27]. Furthermore, the interaction between the electrode materials and the respective conventional spiro-OMeTAD layers can also affect the overall lifetime of the device, which was previously corroborated by many researchers [28,29]. The robust degradation of ETLs, HTLs, and the perovskite crystal structure of light-harvesting materials in PSC devices due to the thermal stress under processing and/or operational conditions were also reported by many researchers [19,30]. In addition, the anomalous behavior of these devices, namely hysteresis, has been highlighted as another significant factor responsible for the unsteady-state performance of these devices [31,32]. The reason behind such PV behavior is still unknown. However, researchers have attributed several factors to it, such as the large dielectric properties of perovskite sensitizers at the nano-scale [33,34], interfacial ferroelectric characteristics of the device [15], ion-migration [35–37], mismatched energy band alignment of the device layers [38,39], the slower trapping–detrapping of charges at the grain boundaries or the barriers of different layers [40,41], inefficient charge transportation or extraction by ETLs and HTLs [42,43], and imbalance in the charge transport through HTLs and ETLs [44,45], etc.

Therefore, in recent years, numerous researchers have tried to overcome the instability issues of PSC devices by introducing numerous strategies against the known responsible factors. Additionally, in some recent works, some researchers have splurged to assemble such strategies in their review articles on PSC devices. Recently, Wang et al. [46] overviewed the factors behind the poor stability of perovskite PVs and discussed the role of some individual promising electron-transporting materials (ETMs), hole-transporting materials

(HTMs), and electrodes for enhancing the stability of such devices. A few years ago, in another study, Asghar et al. [47] assembled all the significant factors behind the poor stability of PSC devices and described all of the experimental methods for evaluating the degradation of these devices as well. However, there was no significant discussion about the strategies of overcoming such issues in those previous studies. The following year, Li et al. [48] provided an extensive overview, specifically on the morphology engineering strategies for enhancing the stability of hybrid PSCs. However, other strategies including the use of interfacial passivation layers between the charge transport layers and the light-harvesting perovskite layer or the suggestion that different ETMs and HTMs be used to enhance the stability of PSCs have been elaborately discussed. Ava et al. [49] specifically assembled the strategies for enhancing the thermal stability of methyl-ammonium lead halide (i.e., $CH_3NH_3PbI_3$)-based PSC devices, although the strategies for increasing the humidity and oxidative stability of these devices were not discussed. In another review work, the specific role of various additives in enhancing the stability of PSCs was elaborately discussed by Liu et al. [50]. However, to the best of our knowledge, there is no such recent work on PSCs wherein all the potential strategies for overcoming the instability issues of all kinds of these devices have been extensively discussed in general, in addition to describing their instability issues in elaboration.

Herein, this review will discuss the exposure of external factors (e.g., heat, moisture, and UV light) that cause PVs instability, and their mechanisms which induce such poor device stability. Furthermore, the potentialities of some highly stable state-of-the-art alternative materials as promising candidates for different device layers such as HTL, ETL and electrodes will be individually and elaborately discussed in the section dedicated to formulating the solution. In a word, this report will help researchers and academic personnel obtain a broad view of the stability issues of PSCs and all the effective strategies to overcome these issues in brief. Additionally, recommendations for further studies are also provided to help researchers further improve this field in the future.

## 2. Stability Problems in Perovskite Solar Cells

### 2.1. Chemical Instability

2.1.1. Instability Due to UV Light Exposure

PSCs show promising stability under the laboratory experimental setup, however, under real-life operation conditions, most of them exhibit poor stability under sunlight and UV illumination [51,52]. It was reported that the presence of oxygen during illumination gives rise to the decomposition of methyl-ammonium perovskite material in the PSC device which acts as a light-harvesting layer. According to the mechanism proposed by N. Aristidou et al., the degradation is first initiated by the atmospheric oxygen molecules capturing free electrons, and thus forming $O_2{}^-$ which leads to the decomposition of methylammonium ions into the methylamine of perovskite structure [53,54]. The degradation reaction is as follows [55]:

$$O_2 \xrightarrow{h\vartheta > E_G} O_2^- \tag{1}$$

$$CH_3NH_3^+ + O_2^- \rightarrow HO_2 + CH_3NH_2 \uparrow \tag{2}$$

$$2HO_2 \rightarrow 2O_2 + H_2 \uparrow \tag{3}$$

In contrast, several studies have reported the severe degradation of the perovskite layer under UV radiation even without atmospheric oxygen. For instance, Salhi et al. [56] outlined a degradation pathway of perovskite materials at the $TiO_2$-based solar cells in the presence of UV light. It is worth noting that the compact or mesoporous Titania ($TiO_2$) are frequently used as photoanodes in PSCs that possess an optimum band gap (i.e., 3.2 eV) to act as a photocatalyst for oxidizing the water molecules present in the atmosphere [56]. Consequently, $\cdot OH^-$ radicals are produced, from which water oxidation ensues, which can further act as a secondary oxidizing agent to extract the electrons from iodide ions present in the perovskite layer. A few years ago, Niu et al. [57] revealed another

pathway for the decomposition of perovskite materials in the presence of $TiO_2$ without implicating atmospheric oxygen and water. According to their preambles, $TiO_2$ can directly extract electrons from $I^-$ and break down the basic structure of perovskite by producing discrete $I_2$ molecules. Additionally, the pertinent organic cations of the perovskite molecule (e.g., methylammoium) are decomposed into independent volatile molecules such as $CH_3NH_2$, leaving behind a number of protons. Finally, the extracted electrons (previously picked by the $TiO_2$ photocatalysts) combine with the protons and produce hydrogen iodides (HIs) that evaporate quickly due to their low boiling point. The entire degradation mechanism can be expressed with the following equations (considering $CH_3NH_3PbI_3$ as the light-harvesting material) [57]:

$$2I^- \leftrightarrow I_2 + 2e^- \tag{4}$$

$$3CH_3NH_3^+ \leftrightarrow 3CH_3NH_2 + 3H^+ \tag{5}$$

$$I^- + I_2 + 3H^+ + 2e^- \leftrightarrow 3HI \tag{6}$$

Therefore, in the presence of moisture or oxygen, or even in their absence of them, UV radiation can cause the severe degradation of the perovskite layers of PSCs which reduces their lifetime and considerably diminishes their PCE.

### 2.1.2. Instability Due to Moisture and $O_2$

The water molecules in the atmosphere can be another important factor in reducing the lifetime stability of PSC devices. They may either directly affect the perovskite layer of such devices by catalyzing the irreversible degradation reactions of the perovskite compounds or may bring up the contemporaneous degradation of the organic–inorganic mixed-perovskite materials associated with the environmental $O_2$ molecules. It is generally known that water can act as an effective catalyst for the irreversible decomposition of perovskite films [58]. Frost et al. [59] reported a moisture-induced decomposition pathway of methylammonium lead iodide perovskite composite, as shown in Figure 2. This shows that the degradation starts with a spontaneous reaction between the water molecule and organic–inorganic mixed-halide perovskite material. Consequently, an intermediate complex compound is formed between these molecules that further results in various discrete volatile and stable compounds such as HI, $PbI_2$, and $CH_3NH_2$.

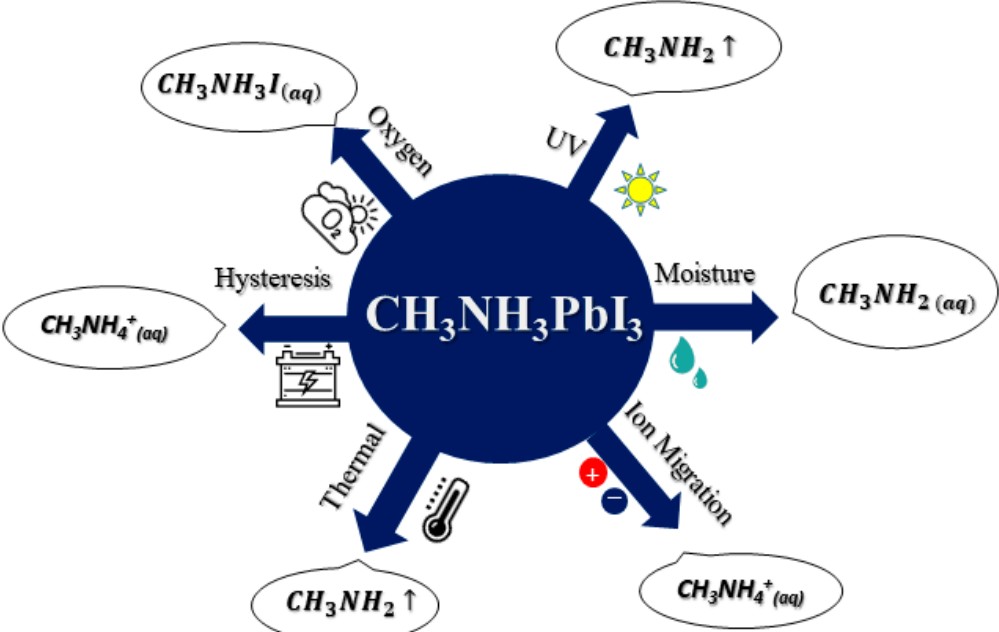

**Figure 2.** Major instability issues of perovskite solar cells.

In another work, Niu et al. [57] outlined some vital factors responsible for the degradation of perovskite films, such as solution processing, UV-light, thermal stress, moisture, and oxygen. Yang et al. [60] experimentally manifested the moisture-induced degradation of perovskite films in their fabricated device by exposing it to different operating conditions with diversified relative humidity. It is noteworthy that perovskite materials are mostly hygroscopic and thereby can be easily hydrolyzed in the presence of moisture. More specifically, the high polarity of a water molecule can affect the hydrogen bonds between the organic and inorganic units of the targeted perovskite material, which further causes the reversible or irreversible decomposition of the material [61]. According to Niu et al. [57], the overall degradation process can be expressed as follows:

$$CH_3NH_3PbI_{3\ (s)} \rightarrow PbI_{2\ (s)} + CH_3NH_3I_{(aq)} \tag{7}$$

$$CH_3NH_3I_{(aq)} \leftrightarrow CH_3NH_{2\ (aq)} + HI_{(aq)} \tag{8}$$

$$4HI_{(aq)} + O_{2\ (g)} \rightarrow 2I_{2\ (g)} + 2H_2O_{(l)} \tag{9}$$

$$2HI_{(aq)} \rightarrow H_{2\ (g)} + I_{2\ (g)} \tag{10}$$

At first, the perovskite material is hydrolyzed in the presence of moisture and turns into two separate molecules, namely $PbI_2$ and $CH_3NH_3I$, as shown in the reaction (7). Later, the as-formed $CH_3NH_3I$ molecules degenerate into relatively more stable compounds (namely $CH_3NH_2$ and HI) through a reversible reaction (8). In this step, the $CH_3NH_3I$, $CH_3NH_2$, and HI molecules co-exist in the perovskite layer in an equilibrium condition. There are two ways for HI degeneration in the next step. One method is a redox reaction in the presence of oxygen, while the other method is a photochemical reaction in which HI can decompose into $H_2$ and $I_2$ under UV radiation. The consumption of HI, according to the reaction, directs the whole degradation process forward and leads to the permanent degradation of the perovskite layer [57,59]. Here, it is worth mentioning that such reactions only occur in the contemporary presence of moisture, oxygen, and continuous UV radiation. However, several pieces of research have evidenced the robust degradation of perovskite layers in PSC devices, even under dark conditions [62,63]. Hence, the presence of moisture and oxygen in the atmosphere can pose a significant threat to the long-term stability of a PSC that may collapse its market demand, regardless of how great the efficiency it provides is.

### 2.1.3. Influence of Ion Migration on Stability

Ion migration is an undesirable labyrinth of PSCs that occurs during the measurement of photocurrent density while scanning it with an externally applied bias and/or during the operation with long-term exposure to continuous light illumination [64,65]. However, temperature-induced ion migration was also reported by many researchers [66,67]. Before starting to discuss the effect of ion migration on device stability, it is more important to know which species (i.e., ions) are drifting and movable in the perovskite structure. Although there are several candidates which may be migrating from their parental perovskite territories into the adjacent interfaces (e.g., HTL/perovskite or perovskite/ETL interfaces), it is mainly the protons or $H^+$ ions, methylammonium ions ($MA^+$) ions, and iodide ($I^-$) ions which migrate from their lattice sites and leave behind a defect state when obtaining a certain amount of energy. Here, it is worth mentioning that an ion will never be able to migrate from its allocated site to the interfaces or any other layers of the device until it obtains threshold energy. The minimum energy required for such ion movement is referred to as the activation energy of ion migration ($E_a$) [68,69]. Mathematically, the term is evaluated by the following Nernst–Einstein equation [70]:

$$\sigma\left(T\right) = \frac{\sigma_0}{T} \exp\left(\frac{-E_a}{KT}\right)$$

where $K$ and $T$ are the Boltzmann's constant and operating temperature, respectively; $\sigma_0$ is a constant, and $E_a$ is derived from the slope of the $[\ln(\sigma T)–1/KT]$ relation. This energy can either be attained by the exposure of perovskite sensitizers to a continuous light illumination or thermal stress, or when an external bias is applied across it [71,72]. The activation energy required for $Pb^{2+}$ ion migration is considerably higher than those of the other drift species, and is rarely attained by light illumination in operating conditions. Therefore, it can be excluded from our discussion.

It is difficult to perceive or exactly characterize how the ions drift from one place to another inside the perovskite device. Therefore, it is usually described by theoretical assumptions based on density functional theory calculations [9–11]. The hopping mechanism is an admissible pathway that has been suggested by many researchers in the literature for ionic transportation in solid materials [73]. Supporting this mechanism and from the outcomes of theoretical density functional theory calculations, Eames et al. [9] suggested a possible migration pathway for $MA^+$, $Pb^{2+}$, and $I^-$ ions in a hybrid lead halide PSC device containing an $MAPbI_3$ sensitizer in its respective layer. The authors propounded that the $MA^+$ ions in perovskite materials migrate by jumping from their allocated position to their nearby defect sites when obtaining threshold energy by applying an external energy source. On the contrary, $I^-$ ions drift through an octahedron in the Pb–I plane, while $Pb^{2+}$ ions move via the diagonal directions. Later, this mechanism was also consented to by other researchers [10,11]. Moreover, in those studies, the Meloni group [11] and Mosconi group [10] postulated a well-described migrating pathway for Pb vacancies, MA vacancies, and iodide interstitials that more explicitly describe the entire ion migration process across the point defects. Additionally, apart from these point defects, the Frankel and Schottky defects, also perovskite crystal lattices in structure, are responsible for the migration of drift ions [73]. The migration of drift ions was also possible due to the local lattice distortions induced by light illumination, piezoelectric effects, accumulated charges, or dissolved impurities in the perovskite crystal structure [74]. Similarly, grain-boundaries are considered another facile migrating pathway for drift ions through the lattice structure of perovskite sensitizers [75,76].

Overall, defects in the crystal structures of perovskite sensitizers are the root cause of such a labyrinth in PSC devices. A series of studies have provided a range of direct or indirect evidence of the delamination or degradation of PSCs due to this undesirable phenomenon [65,77,78]. The most popular model—corroborated by claims from almost all researchers—is that the degradation of a PSC device predominantly originates from the corrosion of the electrode material, i.e., when the drift halide ions, such as iodides ($I^-$), gain the threshold activation energy for their movement, they migrate through the active perovskite material and the adjacent charge transport layers (CTLs) towards the metal cathode, such as silver (Ag) [79,80]. Consequently, the electrode becomes corroded and forms a soluble or insoluble inactive compound, such as silver iodide [81], on the top of the device that inhibits the proper charge extraction through the CTLs and forms an undesirable dipole interface layer [82,83].

Therefore, ion migration is an inevitable intrinsic problem of PSC devices. Although the oxygen, moisture, or UV-mediated degradation of perovskite materials can be eliminated by encapsulation or other effective strategies, ion migration cannot be completely avoided. However, various products of effective device engineering such as interfacial passivation [65,84], the passivation of grain boundaries [85,86], the use of defect-free perovskite materials as light absorbers [87], as well as the use of alternative less-reactive halides in perovskite structure having a comparatively lower activation energy than that of the iodides [88,89] may help to considerably reduce this labyrinth.

### 2.2. Thermal Instability

#### 2.2.1. Thermal Degeneracy of Perovskite or Crystal Structure

Thermal degeneracy is one of the most important issues for perovskite solar cell research [90–92]. High temperatures can easily cause the degradation of the perovskite

crystal structure and phase [93]. The change in the perovskite crystalline phase has a direct effect on the performance as well as the stability of PSC. It has been reported that the thermal effect is a key issue in causing the instability of halide perovskite [94–97]. Conings et al. [98] revealed that $CH_3NH_3PbI_3$ perovskite materials could be decomposed into $PbI_2$ at 85 °C within 24 h. Due to the lower formation energy, halide perovskite rapidly degraded into solid $PbI_2$ and other volatile substances upon heating. Philippe et al. [99] studied the effect of high temperature on $CH_3NH_3PbI_{3-x}Cl_x$ and $CH_3NH_3PbI_3$ perovskite solar cells. They heated the film in an ultra-high vacuum chamber at 100 °C in the absence of air and moisture. The result of their study indicated that $CH_3NH_3PbI_3$ perovskite is not stable at high temperatures (>100 °C).

$$CH_3NH_3PbI_3 \rightarrow CH_3NH_2 + HI + PbI_2 \tag{11}$$

when halide perovskite is heated, the thermal degradation of $CH_3NH_3PbI_3$ and $CH_3NH_3I$ occur under an inert atmosphere proceeds as:

$$CH_3NH_3PbI_3(s) \rightarrow CH_3NH_3I(aq) + PbI_2(s) \tag{12}$$

$$CH_3NH_3I(aq) \rightarrow NH_3 \uparrow + CH_3I(l) \tag{13}$$

Grätzel and coworkers [100] investigated the thermal effect on perovskites and their constituents. They found that the deposition techniques could cause the phase transition of perovskite. The authors observed that a tetragonal to cubic phase transition only occurred on lead iodide precursor-based perovskite, not lead chloride-based perovskite. The phase transition of perovskite solar cells was caused by the fact that the internal temperature of the material raises more than their transition temperature. In the decomposition of perovskite, the loss of $CH_3NH_2$ depends on the hydrogen iodide because of its dense incorporation in the perovskite's matrix. Pisoni et al. [101] observed the very low thermal conductivity of single-crystalline and polycrystalline organometallic perovskite $CH_3NH_3PbI_3$. The stability of the $ABX_3$ (A = alkali; B = Ge, Sn, Pb, and X = halide) perovskite structure is also defined through tolerance factor ($\tau$), which is defined as $(r_A + r_X)/\sqrt{2}(r_B + r_X)$ where $r_A$, $r_B$, and $r_X$ are the ionic radiuses for the ions in the A, B, and X sites, respectively. The values of $\tau$ between 0.8 and 1 represent the most stable perovskite, and the increase in $\tau$ leads to an increase in symmetry [102,103]. Except for the ion radius, phase transition in response to temperature and pressure is also a significant aspect according to the structure stability.

### 2.2.2. Thermal Degeneracy of HTM Layers

HTM is usually obliged to reduce the transporting barrier and block the electron transport between the perovskite and electrode with less carrier recombination to boost the device efficiency. HTMs play a significant role in increasing the efficiency of PSCs within efficient hole extraction and transfer in inverted planer (p–i–n) PSCs [104]. Moreover, HTL plays a crucial role in blocking electrons, transporting holes, and protecting the perovskite from moisture, heat, oxygen, etc. [105]. There are three types of HTMs used in PSCs, namely inorganic, organic, and polymer. In particular, Poly (3, 4-ethylenedioxythiophene) polystyrene sulfonate (PEDOT: PSS) is most commonly used as an HTL in inverted PSC structures. The advantage of the PEDOT: PSS-based HTL is the high conductivity and low cost. On the other hand, this leads to the chemical instability of perovskite solar devices due to the acidic and high hygroscopic nature of PEDOT:PSS [106–108]. Vitoratos et al. [109] reported the thermal degradation mechanism of PEDOT: PSS at 120 °C under environmental conditions. They found a decrease in the electrical conductivity of perovskites with aging due to the shrinkage of PEDOT conductive grains. During thermal aging, the PSS chains are perplexed at the first stage of heating and the ionic bond between PSS chains and PEDOT started to break at the second stage of heating. At the final stage of heating, the X-ray photoelectron spectroscopy revealed that the bond between PEDOT and PSS breakdown.

Despite this instability issue, the oriented arrangement of PEDOT:PSS monolayers provides a higher work function and stronger hydrophobicity, which enhance the $V_{oc}$ and stability of PSCs [110]. On the other hand, oxide-based inorganic HTMs such as NiO with an inverted structure exhibited better air [111,112] and thermal stability [113].

Habisreutinger et al. [114] investigated the thermal stability of most commonly used HTLs, such as: 2,2′,7,-7′-tetrakis($N$,$N$-di-p-methoxyphenylamine)-9,9′-spirobiflfluorene (spiro-OMeTAD), poly(3-hexylthiophene) (P3HT), and poly(triarlyamine) (PTAA) on PSCs. The result of this investigation is shown in Figure 3, and a rapid degradation of the perovskite films was observed for all three HTLs. To overcome this thermal instability issue, organic HTLs were replaced with polymer-functionalized single-walled carbon nanotubes embedded in an insulating polymer of polycarbonate and poly (methyl methacrylate).

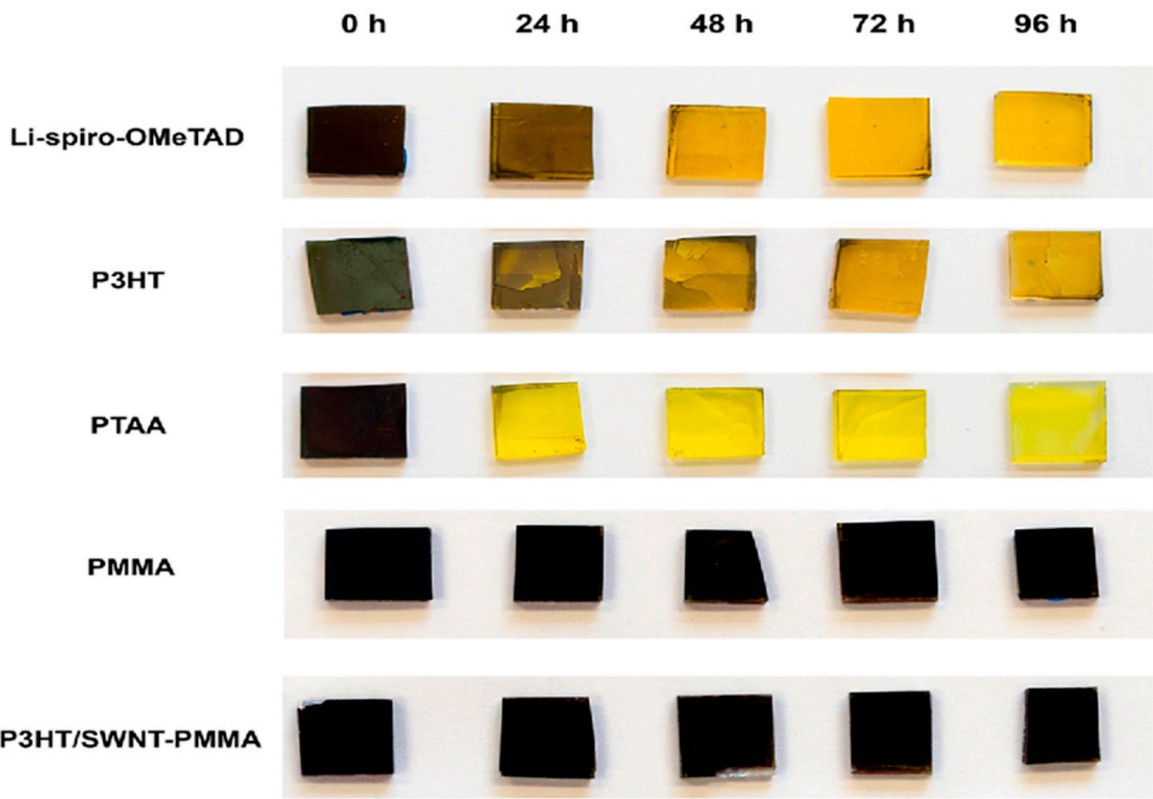

**Figure 3.** Photographs illustrating the visible degradation of the perovskite layer with different hole transport layers exposed to 80 °C under ambient conditions. The conversion from the dark initial color to yellow indicates the degradation of the perovskite material to lead iodide. Over 96 h of heat exposure, only a composite structure of carbon nanotubes and PMMA was able to prevent degradation. Reprinted with permission from Ref. [114]. Copyright: American Chemical Society 2014.

2.2.3. Thermal Degeneracy of ETM Layers

Highly efficient PSCs generally require good electron selective contact between ETM and perovskite materials to reduce the potential barrier for electron transfer while blocking the hole transport to minimize carrier recombination at the interface [115,116]. ETMs play a significant role in increasing the efficiency of PSCs in the extraction of efficient photogenerated electrons from photoactive layers and transport them to the cathode [117]. In addition to the perovskite's thermal stability, ETL also greatly influences the device's significant long-term performance. At present, many types of ETMs such as $TiO_2$, $SnO_2$, and ZnO, as well as some doped oxides are used in n–i–p PSCs [118,119]. The ETL in p–i–n PSCs is usually composed of fullerene or one of its derivatives (for example, [6,6]-phenyl-$C_{61}$-butyric acid methyl ester (PCBM) or indene–$C_{60}$ bisadduct (ICBA)) [120]. The use of organic materials as ETMs offers a low solution processing temperature [121], easy fabrication [120],

and negligible hysteresis [122]. Titanium dioxide ($TiO_2$) is the most frequently used ETL for perovskite solar devices. However, non-stoichiometric defects, such as oxygen vacancies and titanium interstitials, can also form in this layer [123]. Perovskite devices with $TiO_2$ ETL show rapid degradation under illumination. Ahn et al. [124] reported the degradation of the photoconversion efficiency of PSCs. They observed that the degradation of PCE results in the decomposition of the perovskite film at the interface with $TiO_2$. The degradation is initiated at the perovskite/$TiO_2$ interface in a $TiO_2$-based device. They showed that the trapped charges at the interface were responsible for the irreversible degradation of perovskites along grain boundaries [124]. Compact $TiO_2$ ETL can be replaced with $C_{60}$ to enhance the stability of perovskite devices, which was explained later.

$C_{60}$ has been used as an interface modification layer for $TiO_2$ and an electron-accepting layer [12,125]. The $C_{60}$-based device exhibited much more stable performance than the $TiO_2$-based device when annealed at 60 °C for 500 h [12]. On the other hand, Pathak et al. [123] observed that Al-doping in $TiO_2$ reduced the number of trap states and passivated the non-stoichiometric defects of $TiO_2$ to improve the stability of encapsulation devices.

*2.3. Hysteresis Problem of Perovskite*

The performance of a PSC device was evaluated by exposing it to a standard light source and contemporaneously scanning it with an externally applied bias. The obtained scanning curve (i.e., photocurrent density ($J$) vs. voltage ($V$) curve) reveals the flow of the photo-generated current through the external circuit of the device within a certain range of voltage, and thereby, the standard steady-state output of the device can be assumed closely from the curve [126]. According to the theoretical preambles, the resulting scanning curve should be independent of the light illuminating history, scan rate, and direction of the applied bias. However, in several research works, a significant contradiction in the $J$–$V$ curves, i.e., the hysteresis of PSCs, has been observed while sweeping the bias dichotomously (forward and reverse scan) or at different sweep rates. The absolute reason behind this incidence is still unacquainted to the researchers. Therefore, hysteresis is an anomalous behavior of a solar cell that is related to the contradiction in the standard $J$–$V$ curves of the device and is defined by the difference in open circuit voltages ($\Delta V_{oc}$) while applying external dichotomous forward and reverse biases across it to evaluate its PV performance [127,128]. This labyrinth of a PSC device significantly affects its overall performance as well as imparts a negative influence on its long-term stability [129–131]. Dualeh et al. [36] were the first who noticed the hysteresis behavior of PSC devices during the retro scanning of their fabricated cell. Later, many other researchers revealed its manifestation in their devices and tried to find the exact reason behind this problem [132,133].

Initially, it was assumed that the difference in the scan rate of the device was the only reason behind such a contradiction in $J$–$V$ curves. However, it was later corroborated by researchers that the hysteresis index of a solar cell somehow originates from its intrinsic properties, rather than the scan rates or direction of the applied bias. Amongst them, the ferroelectric and large dielectric properties of perovskite sensitizers at the nano-scale [34,134], the interfacial ferroelectric characteristics of the device [18], ion migration [35,36], mismatched energy band alignment of the device layers [38,39], slower trapping–detrapping of charges at the grain boundaries or the barriers of different layers [40,135], inefficient charge transportation or extraction by ETLs and HTLs [42,43], imbalance in the charge transport through HTLs and ETLs [44,45], various structural or chemical changes of the involved materials (e.g., perovskite absorber, ETL, and HTL) during operation or in response to various hostile environmental parameters such as temperature and humidity [134,136], undesirable charge trapping at the defect sites of different layers of the device created by such deteriorations [137,138], etc., are attributed to such anomalous PSC device behavior. Some researchers even mentioned it as a light-induced hysteresis in solar cell devices [35,129,139]. However, the actual reason still could not be assigned. It was assumed that a couple of reasons may act simultaneously to instigate such phenomenon in a PSC device, or sometimes, when the device fulfills all of the requirements to become hysteresis-free, only one reason

among the aforementioned possibilities may also bring up severe hysteresis. Therefore, to eliminate such a problem, we must first identify its origin in the device structure. All of the researchers tried to rationally explain the reasons they attributed for such an anomalous behavior of PSC devices and fortunately, the majority of the reasons were found to be related to the perovskite layer.

For instance, in terms of ferroelectricity and/or large dielectric constant, the hysteresis of PSC devices is directly related to the intrinsic properties of the perovskite layer. Juarez-perez et al. [140] reported the giant dielectric response of $CH_3NH_3PbX_3$ perovskite material at their fabricated device (i.e., FTO/compact-$TiO_2$/$CH_3NH_3PbI_{3-x}Cl_x$/Spiro-OMeTAD/Au) when an external bias was applied across its two terminals. After investigating several samples with different compositions, layer contacts, and internal morphologies, the authors concluded that the giant dielectric constant of the solar device is an intrinsic effect of the used materials rather than interfacial ones. Some other researchers also acquiesced with the result in their studies [141–143]. According to Wasylishen et al. [144] and Mashiyama et al., [145], the tumid freedom of rotation of $CH_3NH_3^+$ cations in a perovskite material can highly facilitate its structural fluctuation when an external field is applied. Thereby, when a PSC device containing $CH_3NH_3^+$ cations in its perovskite layer is scanned with an applied bias in two dichotomous directions, the residing domains fluctuate differently from their actual positions and as a consequence, the flow of charges also changes uncertainly. Thus, the shapes of the *J–V* curves contradict each other [140].

In contrast, inefficient charge extraction in the perovskite layer and unbalanced charge transportation between the ETLs and HTLs of solar cell devices may also give rise to severe hysteresis in PSCs [44,45,146]. The slower trapping–detrapping of charges at the grain boundaries and some unwanted charge trapping incidences at the defect sites created due to various operational and/or environmental instability issues are attributed to these phenomena [135,143,147]. To obstruct such charge trapping in the perovskite interfaces, it is first necessary to point out its origin. It is noteworthy that perovskite absorbers often contain several bulk defects during their fabrication [148–150]. In the presence of light, a large number of photo-excited holes and electrons are generated due to the fast dissociation and low binding energy of the sensitizer [151–153]. The photo-excited charges then accumulate the defect sites to reduce the defect density of the perovskite interfaces and thereby, under the continuous illumination of light, it becomes almost impossible to alleviate the charge trapping at the interfaces of device layers. This means that perovskite sensitizer is the only one that is responsible for all such phenomena with its bulk defect sites. In addition, the mismatched alignment of the energy bands of device layers such as ETL, HTL, as well as the electrode and perovskite layers also renders the hysteresis issue of a solar cell, even though all of the used materials and their compositions support the hysteresis-less behavior of the device [154]. Wu et al. [155] alluded that charge collection in the external circuit is largely affected by the mismatching of energy levels at the perovskite sensitizer/carbon electrode interface which is conceded as one of the reasons behind the hysteresis issue of the solar cells. Most recently, Sun et al. [156] showed that the energy level mismatch at the perovskite/ETL interface renders the accumulation of movable charges and ions that engenders severely anomalous *J–V* hysteresis.

Overall, in conclusion, it is clear that anomalous behavior is an intrinsic problem of PSCs—not an interfacial one. The majority of them originate from the perovskite layer of the device, although the misalignment of energy bands while fabricating the device with different layers may also give rise to a severe hysteresis index. Problems may also arise by using conventional $TiO_2$-based ETL; however, passivating the direct contact between the perovskite sensitizer and ETL with an excess passivation layer may resolve this problem. We will go for a detailed discussion of the strategies for eliminating the hysteresis of PSC devices in Section 3.4.

### 3. Improvement Strategies for the Instability Problems

*3.1. Enhancing Stability of Perovskite Sensitizer*

In the presence of UV-radiation, light, air, moisture, temperature, etc., can cause the severe degradation of the perovskite layers of PSCs that reduces their lifetime. The degradation of PSCs in a humid environment and under high thermal conditions is a challenging issue. Among different factors, moisture and heat have been considered to be among the biggest challenges. Many researchers are concerned with this instability issue and have made many attempts to enhance the stability of perovskite materials in humid and thermal environments. Solving the instability problem of perovskite material is a critical strategy for upgrading their long-term stability.

In a humid environment, the stability of PSCs largely depends on their structural configuration, such as $MX_6^{4-}$ (M denotes metallic cations: $Pb^{2+}$, $Sn^{2+}$, and $Cu^{2+}$; and X denotes anions: $I^-$, $Br^-$, $Cl^-$, and $SCN^-$) or $PbX_6^{4-}$ octahedral structures and the connection between A cations (($CH_3NH_{3+}$, MA), ($NH_2CHNH^{2+}$, FA), $Cs^+$, ($CH_3CH_2NH^{3+}$, EA), or ($N(CH_3)^{4+}$, TMA) and the neighboring eight octahedral [60]. The lattice parameters of perovskites and their photoelectric performance are directly influenced through the sizes of A and X ions [157]. Orthorhombic or tetragonal perovskite can be transformed into cubic phase with symmetric structures after A cation or X anion are substituted with other ions. Therefore, adjusting the elements and composition in perovskite structures can stabilize their structure [158]. The conception of a mixed-halide perovskite can help enhance the humidity stability of PSCs. The cubic $CH_3NH_3Pb(I_{1-x}Br_x)_3$ solid solution perovskite demonstrated more stable humidity compared to the tetragonal $CH_3NH_3PbI_3$ perovskite with 12.3% conversion efficiency (Figure 4a) [157]. The stability of the $CH_3NH_3^+$ cation in the $CH_3NH_3PbI_3$ lattice structure will enhance due to the inclusion of the bromide ion (Figure 4b). As shown in Figure 4a, $CH_3NH_3Pb(I_{1-x}Br_x)_3$ ($x \geq 0.2$) exhibited only a slight degradation after a humidity soaking test exposing 55% relative humidity. Based on the above discussion, it can be assumed that the multiple-halogen perovskites ($MAPb(I_{1-x}Br_x)_3$ and $MAPb(I_{1-x}Cl_x)_3$) might also possess stable moisture-resistant structures.

Furthermore, some researchers demonstrated the long-term stability of their devices by incorporating different organic cations such as $CH_3(CH_2)_3NH_3^+$ and $C_6H_5(CH_2)_2NH_3^+$ into the $CH_3NH_3PbI_3$ matrix. They observed that a two-dimensional (2D) layered perovskite exhibited better moisture stability than the three-dimensional (3D) $CH_3NH_3PbI_3$ perovskite film. These hydrophobic organic cations helped enhance the moisture stability to reduce the loss of $CH_3(CH_2)_3NH_3^+$ and $C_6H_5(CH_2)_2NH_3^+$ cations in a moist atmosphere. Cao et al. [159] reported the solar cell application of a layered $(CH_3(CH_2)_3NH_3)_2(CH_3NH_3)_{n-1}Pb_nI_{3n+1}$ $((BA)_2(MA)_{n-1}Pb_nI_{3n+1})$ perovskite light absorber with enhanced moisture stability. Their long $BA^+$ chains with hydrophobic properties and highly compact 2D perovskite structures can prevent direct contact between water and the perovskite, which would chemically enhance the stability of perovskites. As shown in Figure 5a, the morphologies and colors of the $(CH_3(CH_2)_3NH_3)_2(CH_3NH_3)_{n-1}Pb_nI_{3n+1}$ perovskite film remain unchanged after two months exposure under 40% humidity conditions, while 3D $MAPbI_3$ gradually decomposes to yellow $PbI_2$ due to the regular loss of $MA^+$ cation. On the other hand, Smith et al. [160] reported the 2D structure of layered $(C_6H_5(CH_2)_2NH_3)_2(CH_3NH_3)_2[Pb_3I_{10}]$ $((PEA)_2(MA)_2[Pb_3I_{10}])$ hybrid perovskite, used as an absorber in solar cells with 4.73% conversion efficiency. They observed that a film of $MAPbI_3$ gradually decomposed to yellow $PbI_2$ after 20 days while the structure of $(PEA)_2(MA)_2[Pb_3I_{10}]$ remained unchanged after 46 days' exposure under 52% humidity condition.

In order for the addition of hydrophobic organic cations ($PEA^+$ or $BA^+$) in 2D perovskite to induce greater stability under humid conditions compared to that of 3D $MAPbI_3$, it is proposed that 2D perovskite films $(CH_3(CH_2)_3NH_3)_2(CH_3NH_3)_{n-1}Pb_nI_{3n+1}$ and $(C_6H_5(CH_2)_2NH_3)_2$ $(CH_3NH_3)_2[Pb_3I_{10}]$ exhibited a much better moisture resistance compared with its 3D $MAPbI_3$, which make them more attractive for large-scale industrial implementation.

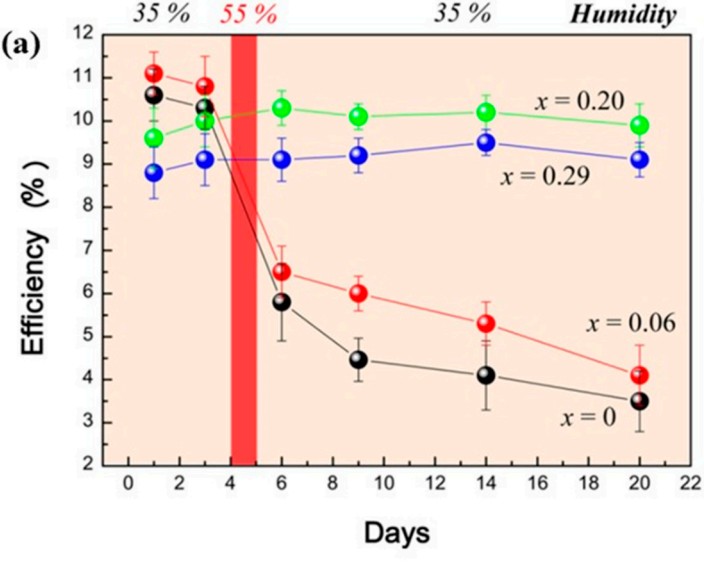

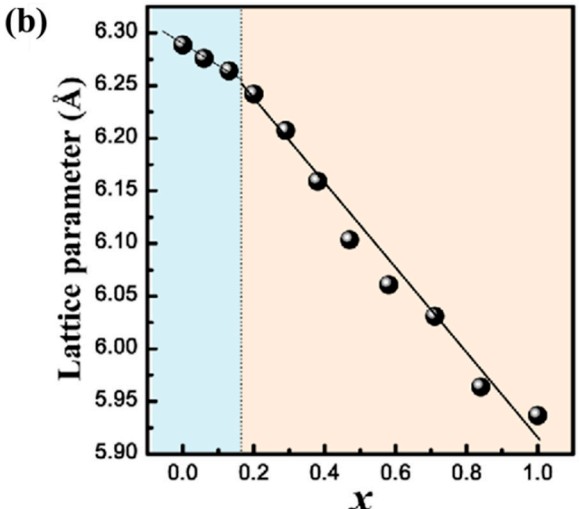

**Figure 4.** (**a**) Power conversion efficiency of the MAPb(I$_{1-x}$Br$_x$)$_3$ (x = 0, 0.06, 0.20, and 0.29)-based solar cells measured in the air (at room temperature) without encapsulation; and (**b**) the lattice parameters of pseudo cubic or cubic MAPb(I$_{1-x}$Br$_x$)$_3$ as a function of Br composition (x). Reprinted with permission from Ref. [157]. Copyright: American Chemical Society 2013.

In recent years, some researchers have used pseudohalide thiocyanate ions (SCN$^-$) to replace some iodides in CH$_3$NH$_3$PbI$_3$ perovskite lattice and proposed a pseudo-orthorhombic structure CH$_3$NH$_3$PbI$_{3-x}$(SCN)$_x$. It was not surprising that CH$_3$NH$_3$PbI$_{3-x}$(SCN)$_x$ solar cells have strong long-term environmental stability because they have excellent moisture resistance during the fabrication process [58]. As shown in Figure 5b, after 500 h of storage in open air with an average RH level above 70%, CH$_3$NH$_3$PbI$_{3-x}$(SCN)$_x$ solar cells without encapsulation retain 86.7% of the initial average PCE. In CH$_3$NH$_3$PbI$_{3-x}$(SCN)$_x$ perovskite film, the interaction between linear-shaped SCN$^-$ and the central Pb$^{2+}$ ion is much stronger and the formation constant of the lead thiocyanate complex (Pb(SCN)$_4$$^{2-}$) is much higher over the PbI$_4$$^{2-}$ complex in CH$_3$NH$_3$PbI$_3$, indicating a stable crystal structure of CH$_3$NH$_3$PbI$_{3-x}$(SCN)$_x$ [161]. The introduction of thiocyanate ions (SCN$^-$) in the MAPbI$_{3-x}$(SCN)$_x$ perovskite crystal lattice largely strengthened the moisture stability compared to the MAPbI$_3$-based solar cell, which makes them more attractive for large-scale industrial implementation. The above review also represents that ion doping might be a viable way to develop the intrinsic stability of perovskite films in a humid environment.

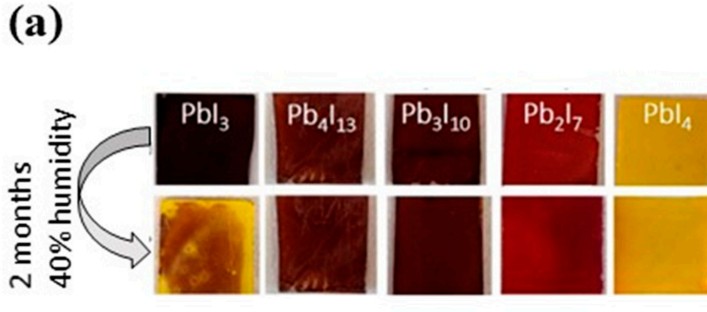

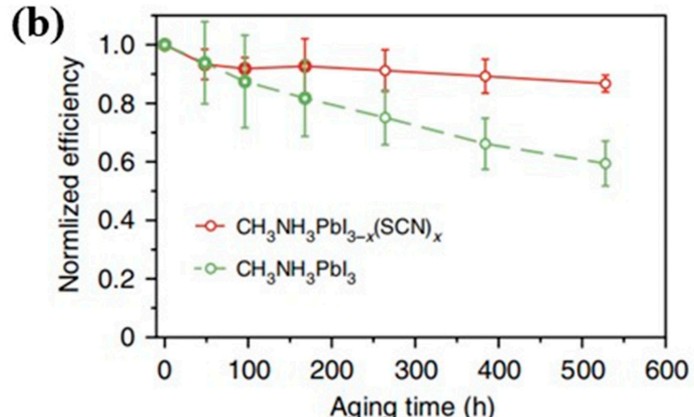

**Figure 5.** (**a**) Images of different $(CH_3(CH_2)_3NH_3)_2(CH_3NH_3)_{n-1}Pb_nI_{3n+1}$ perovskite films before and after exposure to 40% humidity (Reprinted with permission from Ref. [159]. Copyright: American Chemical Society 2015); and (**b**) the evolution of the PCEs of $CH_3NH_3PbI_3$ and $CH_3NH_3PbI_{3-x}$ $(SCN)_x$-based solar cells after gaining in air without encapsulation (Reprinted with permission from Ref. [58]. Copyright: Springer nature 2016).

Apart from the moisture resistance, the thermal stability of PSCs is also raising a lot of concern. Due to its lower formation energy, halide perovskite become rapidly degraded into solid $PbI_2$ and other volatile substances on temperature change. Kim et al. [98] reported that the thermal degradation of halide perovskite is caused by an internal factor linked to materials. Hence, the best option is using thermally resistant materials to prevent thermal decomposition [162]. Many researchers reported that the mix of halide on the perovskite layer or replacing I with Br or Cl, or mixing the cation on-site A of $ABX_3$ (A = $CH_3NH_3$, Cs, B = Pb, Sn and X = I, Cl, Br) perovskite could enhance the thermal stability of halide perovskite [163–165]. The Pb-based PSC has shown a certified efficiency of over 22% [166,167]. However, their poor stability and high toxicity are the main obstacles to their commercialization. The replacement of Pb with less toxic Sn reduced the toxicity problem but the stability problem remained [168]. Liu et al. [169] reported that another divalent metal Pd-based perovskite may represent a better solution against the toxicity and stability issues. They observed that 3D perovskite $MAPdI_3$ and 2D perovskite $(MA)_3Pd_2I_7$ are thermally stable up to 250 °C, comparable to another hybrid perovskite [170]. On the other hand, Zheng et al. [171] fabricated three kinds of mixed dimensional (MD) perovskite using a non-toxic transition metal cation ($Zn^{2+}$, $Mn^{2+}$, and $Ni^{2+}$) to partially replace the Pb cation in $MAPbI_3$. These three MD perovskite devices showed good stability against high humidity and temperature. The MD perovskite devices can retain 84%, 85%, and 76% of their original PCE at 60 °C for 100 h.

In contrast to organic–inorganic $MAPbI_3$ perovskite, inorganic $CsPbI_3$ perovskite exhibited better thermal stability with 9.3% conversion efficiency [172,173]. Despite the better thermal stability of inorganic halide perovskite, they are showing poor performance

due to the small bandgap and high deep-level defect concentration [174]. Thus, mixed halide perovskite or halide perovskite alloy can be a favorable condition for showing better thermal stability with high performance [3].

On the other hand, the substitution of methyl-ammonium, MA($[CH_3(NH_3)]^+$), with formamidinium, FA ($[NH_2(CH)NH_2]^+$), may enhance the thermal stability of halide perovskite [175,176]. The $FAPbI_3$ perovskite exhibited remarkable thermal stability at high temperatures (25–150 °C) [176,177]. Moreover, the mixed cation perovskite $Cs_xFA_{1-x}PbI_3$ exhibited high PCE and good thermal stability [175]. This mixed cation perovskite exhibited no discoloration after 60 min at 150 °C with 15% PCE. Therefore, the mixed cation and halide perovskite showed better performances against thermal decomposition, and their power output reached up to 18~21.1% [49,162,175,178].

### 3.2. Enhancing Intrinsic (Device) Stabilities

3.2.1. Improvement and Modification of Electron Transport Layers (ETLs)

ETLs played an exigent role in the improvement of the PCE of PSCs by facilitating the collection of the photo-excited electrons and, subsequently, their facile transportation to the respective electrodes from the photo-sensitized perovskite layer [143]. Moreover, in heterojunction planar-structured devices, they act as hole-blocking layers (HBLs) that suppress the unwanted transitions of photo-generated holes from the active perovskite layer to the FTO-substrate [179]. In addition, ETLs also act like a scaffold in the mesoporous structured PSCs to support the formation of the active perovskite layer [180–182]. However, several groups of researchers have demonstrated some ETL-free devices but ultimately, they could not outstrip the ETL-incorporated PSCs in terms of stability and efficiency [183]. The key destination of the present research works for ETLs is to obtain an enhanced charge carrier lifetime, high electron conductivity with an improved energy alignment, as well as to obtain a highly stable performance with it. In this context, different approaches have been followed (Figure 6). Each of these strategies is described below.

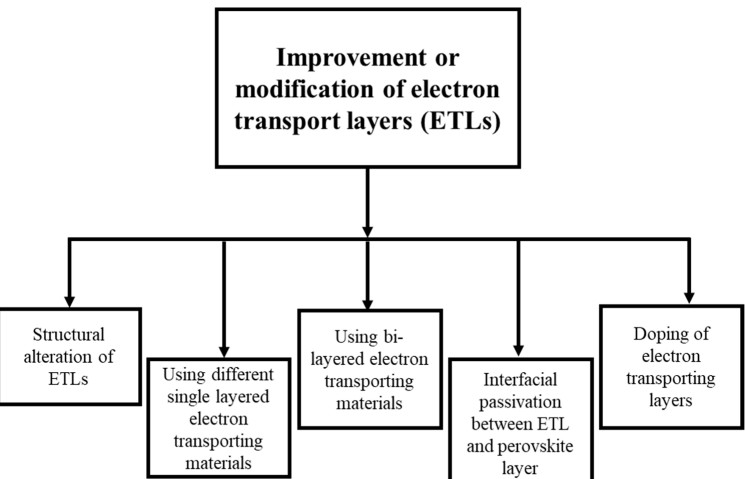

**Figure 6.** Improvement or modification strategies for electron transport layer of PSCs.

Single-Layered Electron Transport Materials (ETMs)

A number of inorganic materials such as metal oxides (e.g., $TiO_2$ [13,14,184], ZnO [185,186], $SnO_2$ [181,182,187], selenides (e.g., CdSe [188]), sulfides (e.g., CdS [143,189]), mp-$Al_2O_3$ [190], fullerene [191], PCBM (i.e., phenyl-$C_{61}$-butyric acid methyl ester) [192], as well as various ternary metal oxides (e.g., $SrTiO_3$ [193], $BaSnO_3$ [194,195], $Zn_2SO_4$ [196,197]) have been introduced by many researchers. However, the majority of the research to date has focused on $TiO_2$-based ETLs, from the very initial stage of its journey due to its low-cost availability and high photo-conversion performance under different experimental conditions.

Each of the proposed materials has some drawbacks in addition to exhibiting various advantages. For instance, several research groups have reported the incitement of perovskite

layer degradation by $TiO_2$ ($E_g$ = 3.2 eV) under continuous light illumination [198,199]. As the $TiO_2$ has a strong capability for extracting electrons from various organic compounds (i.e., perovskite absorbers) as well as negatively charged electrodes, the devices based on $TiO_2$-ETLs exhibit comparatively lower stability compared to other metal oxide or mixed metal oxide-based PSCs under continuous light illumination [200]. Some researchers also mentioned that, due to its excellent photocatalytic property, it can easily oxidize the $H_2O$ (i.e., moisture content) of its surrounding environments and subsequently forms a highly reactive secondary oxidizing agent (i.e., $\cdot OH^-$ radicals) under UV illumination, which further suggests the destructive degradation of the organic perovskite layer [201–203].

To overcome such labyrinths, various alternative ETMs have been proposed by researchers. Among them, $SnO_2$ has gained much attention in recent years as the most promising alternative candidate of $TiO_2$ for the ETLs of perovskite PVs [16,204–206]. The primary reasons behind its popularity in this context may be its deep conduction band [207], wide optical bandgap (i.e., 3.6–4.0 eV) [208], high electron mobility (up to 240 $cm^2/V \cdot s$) [209], excellent UV resistance properties [210], chemical stability [211,212], easy processing by low-temperature methods (e.g., at less than 200 °C) [89,213], and finally, its lower photocatalytic activity as compared to the conventional $TiO_2$ or other metal oxide-based ETLs [204]. Furthermore, ZnO has also been reported as a potential candidate for ETLs due to its excellent electron mobility (i.e., 205–300 $cm^2/V \cdot s$) and high electron-conducting properties [214,215]. However, in terms of performance and stability, it still lags behind conventional $TiO_2$-based ETLs. A few years ago, Wang et al. [216] revealed that ZnO can easily interact with the organic perovskite compounds when the device is annealed at high temperature (i.e., >100 °C). This chemical interaction reduces the stability of the cell. In another recent work, Mali et al. [217] showed the robust thermal degradation of ZnO-based ETLs in their thermal stability analysis. Ma et al. [15] showed that the long-term instability of ZnO-ETL-based PSCs can be a consequence of chemical incompatibility between organic–inorganic perovskite absorber and ZnO at their interface. Similar results were also reported by many other researchers that proved the incompetence of single ZnO-based ETLs for the fabrication of stable PSC devices as compared to the other proposed metal oxide ETLs [218–220]. However, in some recent works, the modification of ZnO-based ETLs with different passivators (at the perovskite/ETL interface to enhance their chemical compatibility) or additives, such as graphene [221], ZnS [214], aluminum [219], $TiO_x$ [222], PCBM [223], and protonated ethanolamine/MgO [224] unraveled some satisfactory results in the long-term stability of PSC devices.

On the other hand, a group of researchers demonstrated the improved lifetime stability of their devices by using mixed ETMs at the respective layer instead of a single metal oxide. For instance, Shin et al. [196] fabricated a device by a solution process at low temperature (i.e., <100 °C) using a ternary n-type semiconductor, i.e., $Zn_2SnO_4$ (ZSO) as ETL, poly-(triarylamine) as HTL, and $CH_3NH_3PbI_3$ as the perovskite absorber. The device demonstrated excellent light harvesting and subsequently, a high electron collecting performance with a moderate PCE (14.85% under experimental AM 1.5G 100 $mW/cm^2$ illumination). The most contrasting aspect of $Zn_2SnO_4$ is its excellent chemical stability in polar organic solvents and various acid-based solutions with a high electron hall mobility (i.e., 10–30 $cm^2/V \cdot s$) [225] which makes it very promising for making ETLs in PSC devices. Similarly, Zhu et al. [226] used a La-doped $BaSnO_3$ (LBSO)-ETL to fabricate a mesoporous PSC and compared its stability with a novel mp-$TiO_2$-based device under continuous light illumination. With moderate PV performance (highest obtained PCE 15.1%), the device exhibited better stability than the reference mp-$TiO_2$-based device. The higher steady-state performance was attributed to the less photocatalytic activity of LBSO than the $TiO_2$.

Bi-Layered ETMs

The use of potential bi-layer interface structures instead of single ETLs has become of significant interest in recent years due to the long-term stability of the devices. Zhang and his colleagues [121] used a ZnO/PCBM bi-layer to fabricate an efficient PSC device with

long-term stability. In addition to being sufficiently efficient in terms of PV performance (e.g., 16.8% PCE), the device showed enhanced stability at room temperature and ambient air (RH ~20–50%). After two months of storage under such conditions, only a 5% reduction in PCE was reported. According to the authors' conjecture, the ZnO interlayer enhanced the device stability by resisting the metal ion diffusion from the outermost electrode to the inner layers as well as by acting as a protecting barrier of the device against moisture and oxygen, while the PCBM layer induced a fast charge extraction tendency. Later, this assertion was justified by many other researchers through their devices. For instance, Lim et al. [227] introduced a polyethyleneimine-ethoxylated layer between the ZnO-based ETL and the perovskite ($CH_3NH_3PbI_3$) layer of their device and obtained 15.8% PCE with long-term performance stability through it. Kown et al. [228] introduced a layer of ZnO nanoparticles between the metal electrode and the PCBM-ETL of their device. The bilayer structure (ZnO/PCBM) contemporarily decreased the charge recombination at the perovskite/ETL interface and significantly reduced the device degradation. The application of the ZnO/PCBM bilayer structure as ETL was also reported by Jiang et al. [229]. The bilayer-based devices showed enhanced PCE (up to 17.2%) compared to the control cell structure (i.e., ITO/PTAA/$MAPbI_3$/PCBM/Al). Furthermore, the double-ETL based device retained almost 95% of the initial PCE after 120 days of storage under dark conditions (RH = 20%), while the control device died within only 10 days. The improved water-blocking ability of the bilayer-modified devices was attributed to the achievement of such a tremendous enhancement in device stability. Deng et al. [230] applied an HI-modified $TiO_2$/$SnO_2$ bilayer for their fabricated device's rapid and efficient electron transportation, leading to 16.74% PCE in actual operating conditions. Intriguingly, the device retained almost 85% of its initial PCE even after heating at 100 °C for 22 h and 91% efficiency after 1400 h of storage under ambient conditions. Wu et al. [219] inserted an aluminum-doped ZnO (AZO) layer between the Zno-based ETL and the perovskite layer of their fabricated device (cell configuration: FTO/ZnO/AZO/$CH_3NH_3PbI_3$/Spiro-OMeTAD/$MoO_3$/Ag). The ZnO/AZO-ETL-based device retained its 91% PCE even after 45 days of storage under dark conditions (a glovebox was used for cell storage with 0.1 ppm water and 30 ppm oxygen concentration), while a 53% drop in the initial PCE was observed within only 12 days for the control device. It was assumed from the XPS analysis that the thin film of AZO passivates the penetration of oxygen and water molecules into the core structure of the perovskite layer, thus significantly enhancing its durability. An outstanding stability in PV performance was achieved using an electron transport bilayer of $C_{60}$/ultrathin-$TiO_x$ in the conventional cell structure of PSCs (i.e., ITO/ETL/perovskite/Spiro-OMeTAD/Au) [180]. After 1000 h of exposure to ambient air conditions and 312 h of continuous UV-illumination, the device retained 90% and 83% of its initial PCE. Hence, using bilayered ETLs can be an effective strategy for enhancing the stability of PCEs.

Doping of ETLs

In addition to the extreme photocatalytic activity of $TiO_2$, sometimes its large surface defect densities may also reduce the stability and performance of the device by inducing poor electron mobility at the charge transport layers. It is evident that the mesoporous structured $TiO_2$ films demonstrate a higher defect density than the single crystals [82,195]. Thereby, a bulk number of charges become trapped in the defect sites and significantly affect the device performance and stability. An alternative strategy, i.e., the doping of conventional ETLs and the use of various self-assembled monolayer molecules (SAM), have been proposed by researchers to resolve this problem. Hou et al. [231] reported an enhanced operational stability of their device (under one sun illumination) by using a fullerene-based ETL and tantalum-doped WOx as a small interfacial layer between the different layers of the conjugated polymers. In addition to possessing a high-PCE (21.2%), the PSC maintained almost 95% of its performance, even after 1000 h of its operation. Similarly, Chavan et al. [232] demonstrated the doping effects of Ruthenium ion ($Ru^{4+}$) in the conventional mp-$TiO_2$-based ETL, while using a mixed perovskite com-

pound, i.e., $(FAPbI_3)_{0.85}(MAPbBr_3)_{0.15}$ in the perovskite layer. The uniform, pinhole-free Ru-doped ETL exhibited long-term stability with negligible hysteresis effect. Under experimental one sun illumination (30–35% RH, temperature 25 °C), the Ru-TiO$_2$-based PSCs retained above 80% of their initial PCE even after 90 days while the PCE of the reference non-doped PSCs were drastically reduced within only 50 days. Using AZO-based ETL, Zhao et al. [233] obtained excellent PV stability. The device retained almost 88% of its initial PCE after 144 h of storage under ambient conditions. However, only 12.6% PCE was obtained with that device. In another study, Spalla et al. [234] used AZO-based ETLs in their fabricated device (i.e., ITO/AZO/MAPbI$_{3-x}$Cl$_x$/P3HT/Au) and demonstrated approximately 75% PCE retention capacity after 1000 h of storage in inert N$_2$ atmosphere. Adding an extra layer of PCBM between the AZO layer and the perovskite layer could also reduce the hysteresis of PSCs, which was reported in another study by Dong et al. [235]. The fabricated device (ITO/AZO/PCBM/perovskite/Spiro-MeOTAD/MoO$_x$/Al) achieved up to 17% PCE with enhanced thermal stability. Mahmood et al. [236] showed that indium doping in the nanofibril structure of ZnO-based ETLs could be an effective way to improve the stability as well as the PCE of PSCs. The rapid transportation of electrons through the nanofibers and the high porosity of materials leads to a highly efficient PSC showing up to 17.18% PCE. Furthermore, when the ETLs were coated with a thin polymer layer, namely polyethyleneimine (PEI), the efficiency increased up to 18.69%. This PEI-coated indium-doped ZnO-based devices showed better PV stability. In particular, over 94% of the initial PCE remained, even after 500 h of storage under ambient conditions without encapsulation.

Interfacial Passivation between ETL and the Perovskite Layer

It has been reported that the interaction between the ETLs and perovskite layers significantly reduces the stability of PSCs [237,238]. Therefore, the inhibition of direct contact between the charge transport layer and the respective perovskite sensitizer by passivation of the typical metal oxide films by various interfacial organic materials can be another fruitful strategy towards the better device stability of PSC [239,240], while Qiu et al. [241] introduced an interfacial polystyrene layer between the TiO$_2$-ETL and the perovskite absorber (i.e., MAPbI$_3$) to alleviate the rapid decomposition of the device. Interestingly, the excess layer demonstrated a four times longer T$_{80}$ lifetime (250 h) than that of the pristine TiO$_2$-based devices inside a dry box (relative humidity 5%). Duan et al. [242] fabricated a highly efficient (PCE 19.02%) and steady-state mesoporous device by introducing a p-type organic material, i.e., poly-(9-vinylcarbazole) between the perovskite layer and the respective mp-TiO$_2$-based ETL. The passivation of the ETL with such an ultrathin small organic layer inhibited the hole-carrier transportation into the sensitizer–ETL interface, and consequently, tremendously improved the charge-extraction abilities, stability, as well as the overall performance of the device. Moreover, this improved the crystallinity of the perovskite layer and significantly reduced the pinholes, which facilitated the steady-state performance of the device during a long time period. Very recently, Wang et al. [25] demonstrated an interfacial layer made of thioacetamide (TAA) to passivate the direct contact between the mixed cationic perovskite layer (i.e., $(FAPbI_3)_{1-x}(MAPbI_3)_x$) and electron transport TiO$_2$ layer of their fabricated device. Compared to the pristine cell structure, the TAA-based devices showed an improved PCE (i.e., increased from 17.65% to 19.14%) due to the rapid charge transfer induction. Intriguingly, the reduced trap density in the perovskite layer by the passivation of TAA reduced the non-radiative charge recombination in the perovskite/ETL interface, hence reducing hysteresis. Li et al. [243] prepared a NaBr-coated TiO$_2$-based cell (i.e., ITO/TiO$_2$/NaBr/perovskite/HTL/Au) that achieved up to 21.16% PCE. Introducing the NaBr-passivation layer between the perovskite and TiO$_2$ layer significantly reduced the hysteresis index of the device (*viz.,* from 0.135 to 0.025). Zhang et al. [244] showed that the insertion of a thiophene layer in the SnO$_2$/MAPbI$_3$ interface could reduce the hydrophilicity of the SnO$_2$ surface, making the device more resistant against degradation under humid conditions. The research group obtained over 20% PCE after introducing this interfacial layer while the pristine cells (without the thiophene layer) could only achieve 17.54%

PCE. The effect of interfacial passivation with alkaline metal fluorides on cell stability was examined in another study [245]. The potassium fluoride passivated fullerene-based device showed only 10% efficiency attenuation after 500 h exposure to ambient conditions (40% RH) while the control device (i.e., $ITO/PEDOT:PSS/MAPbI_3/C_{60}/BCP/Ag$) lost over 20% efficiency under the same condition. Tian et al. [46] applied an aminofunctionalized polymer, namely PN4N, as an interlayer between the $CsPbI_2Br$ perovskite and $SnO_2$-based ETL. This time, less than 10% attenuation of PCE was observed with the device after 400 h continuous exposure illumination equivalent to that of to one sun. Therefore, interlayer passivation can be an effective way of enhancing the device stability, reducing hysteresis, as well as improving the PV performance of PSCs.

Structural Alteration of ETLs

The structure of the ETL also plays a vital role in device stability. Qiu et al. [241] showed that, the crystalline anatase-$TiO_2$-based devices exhibit comparatively higher stability than the amorphous-$TiO_2$-based devices. On the other hand, the simple planar structured devices demonstrated better stability when operating without a compact $TiO_2$ layer. However, the efficiency was significantly compromised in that case. In another study, Fakharuddin et al. [246] reported the better stability of the mesoporous $TiO_2$ and $TiO_2$-nanorod-based devices than the planar counterparts. The author claimed that the discontinuous voids originated throughout the perovskite layer due to its time-flowing decompositions, thwarting the constant transportation of photo-generated charges. Consequently, the PCE drops within a very short time, and a significant operational stability reduction is introduced. Nanorod and mesoporous structured ETLs facilitate the continuous charge transportation by maintaining the discontinuous channels created by such phenomena, and thereby help to considerably enhance the device stability. Recently, Karavioti et al. [247] demonstrated a highly stable HTM-free carbon-based device with a mesoporous $TiO_2/ZrO_2$ scaffold. The mesoscopic $LEU_xMA_{1-x}PbI_3$ perovskite sensitizer-based devices maintained almost 100% of their overall performance under experimental conditions (i.e., RH $-60\%$, room temperature) even after 120 days. Likewise, towards the target of obtaining better device stability, mesoporous structured ETLs have been reported so many times in recent years that the question arises now whether the use of mesoporous structured ETLs will be crucial for the long-term stability of PSC devices in the future.

3.2.2. Improvement and Modification of Hole Transport Layers (HTLs)

HTL is one of the basic segments of a PSC device structure that significantly contributes to attaining a high PV performance. The main function of this layer is to collect the photo-generated holes from the perovskite layer and transport them towards the counter electrode, i.e., the cathode layer of the device. Therefore, proper energy band alignment with the adjacent perovskite layer, good charge conductivity, and adequate charge mobility should be prioritized while selecting HTMs for a PSC device. In this context, an organic–inorganic hybrid state-of-the-art material, e.g., 2,2′,7,7′-tetrakis-(*N,N*-di-p-methoxy-phenyl-amine)-9,9′-spiro-bifluorene (spiro-OMeTAD) has mainly been used so far in different research works due to its enhanced PV performance, amorphous nature, and optimal solubility in various organic solvents [51,60,162,248–250]. Furthermore, poly(tri-arylamine, a polymeric organic material, has been introduced by many researchers to yield high certified PCEs in regular planar structured devices [155,167,251–253]. However, in terms of stability, both materials have failed to gratify the researchers due to their various intrinsic properties. A few years ago, Hawash et al. [254] reported the presence of a bulk number of pinholes in the spin-coated spiro-OMeTAD HTL of their fabricated device when exposed to ambient air conditions for 24 h [255]. Jung et al. [27] propounded that such pinholes in HTLs facilitate the inward diffusion of gaseous $O_2$ and $H_2O$ molecules from the ambient air of the surrounding environment when a device is exposed to its operating conditions. In addition, some researchers claimed that these pinholes may also facilitate the outward diffusion of internal compounds/elements having high vapor pressure such as $I_2$, HI, $CH_3I$, $CH_3NH_2$,

NH$_3$, etc., which had been previously formed in the reversible decomposition reactions of the perovskite sensitizer in the presence of light, moisture, or extreme temperature effects [18,96,255]. Thereby, the overall device undergoes robust degradation within a short-time period when such conventional HTLs are used.

Therefore, to enhance the stability of the HTLs of a device, we arrayed all of the strategies that have been proposed to date in literature into three major sections. These are (*i*) using non-hygroscopic dopants or additives that help diminish the rate of decomposition of the conventional HTLs (e.g., PTAA, spiro-OMeTAD); (*ii*) using alternative HTMs instead of the conventional materials in the respective layer; and (*iii*) using HTL-free devices.

Using Non-Hygroscopic Additives

As we stated above, the hygroscopic additives of HTLs such as lithium-bis(trifluorome thanesulfonyl)imide (say, Li-TFSI) with 4-tert-butylpyridine (TBP) can bring off the accelerated degradation of the respective device due to the rapid hydration and aggregation of the additives. Therefore, some alternative or auxiliary additives for this duo were proposed in the literature to enhance device stability and obtain a steady-state performance. Recently, Pham and his colleagues [256] proposed the usage of alkaline-earth bis-(trifluoromethanesulfonyl)-imides (e.g., Ca-TFSI$_2$ and Mg-TFSI$_2$) instead of the conventional Li-TFSI additives for enhancing the efficiency of the device. The comparatively more hydrophobic nature of these additives exhibited considerable resistance against the moisture-induced degradation of the devices as well as effectively stabilized the as-formed co-ordination complexes between the TBP and TFSI-salts. Furthermore, the addition of such alkaline-earth-based additives improved the charge mobility of HTLs and provided a better interface between the HTLs and the light-harvesting layer, which facilitated the charge extraction process of the HTLs. As a result, an overall high performance of the device was obtained (i.e., over 20%) with enhanced stability. In another study, Zhou et al. [257] reported a 2D graphitic N-rich porous carbon (NPC) metal–organic framework as an auxiliary additive to the Li-TFSI/TBP-doped HTLs that enhanced the ultimate long-term stability of the device performance. The hydrophobic nature of the auxiliary additive reduced the aggregation of Li-salts with the TBP and enhanced resistivity against moisture degradation. Additionally, the inherent porosity and the hydrophobic nature of NPC restricted the Li$^{2+}$-permeation into the anode layer, which improved the stability and workability of the electrodes for a long-lasting period. Consequently, the device exhibited a better-stabilized performance than the conventional Li-TFSI/TBP-doped devices (i.e., maximum PCE = 18.51%, fill factor 76%, and over 85% of the initial performance was maintained even after 720 h of exposure to ambient air). Liu et al. [258] propounded a PbI$_2$-modified TFSI/TBP-doped HTL to improve and stabilize the performance of the pristine device structure. The strong interaction between the TBP additives and the PbI$_2$ auxiliary additives inhibited the evaporation of TBP. As a consequence, the agglomeration of Li-TFSI salts was considerably reduced, leading to the lower formation of voids/pinholes throughout the HTL and a better steady-state PCE response of the device. Caliò et al. [259] proposed another N-heterocyclic p-type hydrophobic liquid material, i.e., 1-butyl-3-pyridinium bis(trifluoromethylsulfonyl)imide as an alternative additive for conventional spiro-OMeTAD HTLs. In addition to enhancing the charge conductivity, the additive reduced the common problem of pinhole-formation in the HTL of the device and consequently reduced the charge recombination as well. Such an outstanding resolution of these problems resulted in a competitive device performance (i.e., maximum PCE of the champion device equal to 14.96%) with enhanced air stability. However, in terms of efficiency, it still lags behind conventional TFSI/TBP/spiro-OMeTAD trio HTL-based devices. A few years ago, in 2017, Pellaroque et al. [260] replaced the Li-TFSI salts with molybdenum tris(1-trifluoroacetyl)-2-(thrifluoromethyl)ethane-1,2-dithiolene and molybdenum tris(1-(methoxycarbonyl)-2-(trifluoromethyl)ethane-1,2-dithiolene as alternative dopants for HTLs that showed better device stability even at a higher temperature than the ambient air conditions (85 °C). The following year, Luo et al. [48] reported a fluorine-containing Lewis acid dopant as an effective substitute for TFSI/TBP additives

for PTAA-polymeric HTL that not only improved the device efficiency but also induced long-term device stability. Likewise, many other alternative dopants such as acid additives (e.g., phosphoric acid) [261], Ag-TFSI [61], spiro-(TFSI)$_2$ [155], copper salts (e.g., CuI or CuSCN) [25,262], and F4-TCNQ [263,264] were proposed in several research works as promising candidates to improve the overall performance and long-term stability of PSCs.

Alternative HTLs

Another effective strategy for improving the device stability is using alternative HTLs instead of the conventional spiro-OMeTAD and PTAA-based HTLs. For a better understanding, we arrayed all of the HTMs that have been reported in the literature to date to enhance the device stability of PSCs, into two major classes. These are (i) alternative conductive organic polymers; and (ii) conductive inorganic compounds. It is worth mentioning that, while selecting an alternative material for hole transport purposes, it must be kept in mind that the material for the devices should be compatible with its surrounding layers in terms of energy-level alignment, as is the case for the conventional spiro-OMeTAD and PTAA HTMs. Furthermore, it should provide competitive charge mobility and conductivity as conventional HTMs do, because it is also necessary for a PV device to attain a promising PCE in addition to providing only a steady-state performance with long-term device stability to be commercialized in the market and to be well-accepted by the consumers.

At the initial age of PSCs, Zheng et al. [149] proposed an oligothiophene derivative, i.e., DR3TBDTT as an alternative HTL for PSC devices. The oligothiophene material showed a well-matched energy level alignment with the employed mixed halide perovskite layer (i.e., $CH_3NH_3I_{3-x}Cl_x$) of their fabricated device. Moreover, the highly hydrophobic nature of HTM retarded the moisture-induced degradation of the device and, consequently, exhibited better stability than that of the conventional spiro-OMeTAD HTL-based devices. However, the device's overall performance reached up to only 8.8% with this configuration. Liu et al. [61] demonstrated a dopant-free tetrathiafulvalene derivative (say, TTF-1) as HTL at their fabricated device that exhibited almost two-fold improved stability than the conventional spiro-OMeTAD-HTL-based devices. Additionally, this time, the device configuration attained over 11% PCE. However, it is quite low compared to the present devices in the market. A few years later, Xu et al. [265] reported a low-cost spiro-(fluorene-9,9′-xanthene)-based three-dimensional oligomeric HTM (namely, x55) that exhibited better device stability than that of the state-of-the-art spiro-OMeTAD-based HTL. Furthermore, due to having a higher hole conductivity, mobility, and a comparatively deeper HOMO-level, the proposed HTL brought off a promising PCE (i.e., up to 20.8%). Recently, Christians et al. [266] fabricated a highly stable device using a mixed perovskite (i.e., $(FA_{0.76}MA_{0.21}Cs_{0.03})_{0.67}Pb(I_{0.89}Br_{0.11})_{0.56}$, stoichiometry determined by XPS measurements) and 9-(2-ethylhexyl)-*N,N,N,N*-tetrakis(4-methoxyphenyl)-9H-carbazole-2,7-diamine (say, EH44) as HTL. The device maintained 94% of its peak efficacy even after 1000 h of the combined stress of moisture, oxygen, and light (including UV lights). Similarly, many other organic polymeric materials such as dual-functional polyaniline [267], poly(3-hexylthiophene)/carbon nanotube/poly(methylmethacrylate) [114], and poly [2,5-bis(2-decyldodecyl)pyrrolo [3,4-c]pyrrole-1,4(2H,5H)-dione-(E)-1,2-di(2,2′-bithiophen-5-yl)ethene] [268] were also reported by researchers as promising alternatives of the conventional unstable HTLs in the literature. Furthermore, some inorganic p-type (i.e., hole-conductive) materials were also reported by many researchers that exhibit well-matched energy level alignment with the conventional perovskite materials, good hole mobility, conductivity, as well as better stability against moisture, oxygen, as is the case for other environmental parameters and the conventional polymeric HTMs. Additionally, when used in practical PSC-devices, such materials, including CuSCN [200], reduced graphene oxide-mixed CuSCN [269], CuI [270], $Cu_2O$ [271], $Ti(Nb))_x$ [272] etc., and attained a promising PCE along with the other well-aligned device components. Therefore, to fabricate a long-term stable device, such materials can also be considered promising candidates instead of conventional HTLs.

HTL-Free Devices

Some researchers endeavored to enhance the device stability by eliminating hole conducting layers from the device structure. Initially, such an HTL-free perovskite PV structure was proposed by Etgar et al. [273]. They used a simple $CH_3NH_3PbI_3/TiO_2$ heterojunction structure by facile equimolar solution, dropping perovskite onto the ETL surface. However, as the very initial device in this configuration, it attained only 7.3% PCE under experimental operating conditions (i.e., light intensity 1000 W/m$^2$, AM 1.5G). In 2014, Zhou et al. [274] demonstrated a solution-processed $CH_3NH_3PbI_3/TiO_2$ heterojunction structured device with a low-cost carbon electrode and achieved over 9.0% PCE. Furthermore, under dark conditions, the device exhibited promising stability (i.e., over 2000 h) without encapsulation. The same year, Mei et al. [275] fabricated a highly stable HTL-free device using a mesoscopic $TiO_2/ZrO_2/C$- triple-layer scaffold and obtained almost 13% PCE. The device showed a steady-state performance over 1000 h under ambient air conditions and under continuous sunlight illumination. Priyadarshi et al. [276] demonstrated a monolithic perovskite module that attained up to 10.74% PV efficiency with high stability. The device maintained 95% of its initial peak efficiency, even after 2000 h of exposure due to operating conditions. In another contemporary study, Chen et al. [277] demonstrated a stable HTL-free device using a mixed anion perovskite (i.e., $CH_3NH_3PbI_{3-x}(BF_4)_x$) that attained over 13% PCE. Later on, Zhang et al. [80] reported an HTL-free device based on $SrCl_2$-modified $CH_3NH_3PbI_3$ perovskite sensitizer that approached up to 16% PCE with enhanced stability. Other than these precedents, many more researchers have reported HTL-free device structures that exhibited comparatively better stability than HTL-comprising devices [127,135,214,278–282]. Another advantageous aspect of these devices is their low-cost fabrication process since one of the basic layers is eliminated in such construction. However, the major limitation of these devices was that they never could outstrip the performance of HTL-comprising devices. Therefore, using HTL-free devices can be an effective solution strategy to obtain a better lifetime, if PCE is compromised a little bit while being commercializing for the marketplace.

### 3.2.3. Improvement of Electrode Materials

In general, electrodes are conjoined with the charge transport layers of a PSC device that comprises the outermost splodge of the whole cell. Since they are the closest layer to the environment, the stability of the electrode materials is significantly important for the long-term operation of the PSC [283]. Silver (Ag) and gold (Au) are the most commonly used electrode material for PSC. Silver electrodes react with iodide ($I^-$) to form silver iodides and gold can diffuse in the perovskites to cause irreversible device degradation. It was proposed that the diffusion of iodine and other volatile components into the Ag layer is accelerated by the exposure to moisture that produces pinholes in the spiro-OMeTAD layer and finally, AgI is formed. The formation of metal iodides on the surface of the electrode causes the degradation of the perovskite layer [29]. Even noble elements including gold (Au) were found to take part in this type of degradation [82,284]. The replacement of Ag or Au electrodes by $Cr_2O_3/Cr$ can be a promising strategy for achieving greater device stability due to their chemical inertness towards iodide or the other used halides [82]. Furthermore, Cr electrodes have been reported to reduce the amount of hysteresis at the metal/MAPbBr$_3$ interface by increasing the interfacial resistance [285]. Mei et al. [286] investigated the thick carbon electrodes and produced an efficiency as high as 12.8%, and the devices showed good stability under 1000 h of light soaking. In some architectures, the carbon cathode devices do not even require a selective p-type contact [287]. The application of a carbon electrode has made the device preparation easier via screen printing and assured exceptional long-term stability (>1000 h) under light and ambient conditions [264]. The Cu electrode in an inverted PSC architecture (ITO/PEDOT/MAPbI$_3$/Cu) has been reported to keep the device (efficiency > 20%) without CuI formation at the MAPbI$_3$/Cu interface even after the prolonged annealing of the device at 80 °C [286].

Dai et al. [287] reported that an overall PCE of 11.1% could be obtained from front illumination, and ~70% of the original PCE could be maintained when illuminated from the

transparent electrode side using a spin-coated silver nanowire (AgNW) transparent metallic electrode. Moreover, some excellent device stability was reported in the literature using double-layer (e.g., $Cr_2O_3$/Cr, $MoO_x$/Al, ZnO/Al) and triple-layer transparent electrodes (e.g., Ni/Au/Cu) as well [288–290].

*3.3. Strategies for Mitigating Hysteresis Problem*

It is clear from our previous discussions (Section 2.3) that hysteresis is an intrinsic problem of PSC devices and originates from the perovskite layer of the device. However, in some cases, the misalignment of energy bands corresponding to the device layers can also render such anomalous behavior in PSC devices [291]. Therefore, it is hard to get rid of this problem only by bringing off external modifications in the device structure rather than changing its intrinsic properties. To date, researchers have suggested some effective strategies that significantly reduced or even eliminated device hysteresis in some cases. Among them, strategies include making defect-free perovskites for the respective layer [138], balancing the hole–electron transport flux across the respective charge transport layers during operation [292,293], and using the good hole and electron-conducting materials that suppress unwanted charge accumulations [294,295] and ensuring proper energy band alignment of the device layers [271,291,296] were suggested as the key strategies to obtain hysteresis-free devices. A summary of the strategies for mitigating the hysteresis problem of PSCs is listed in Figure 7.

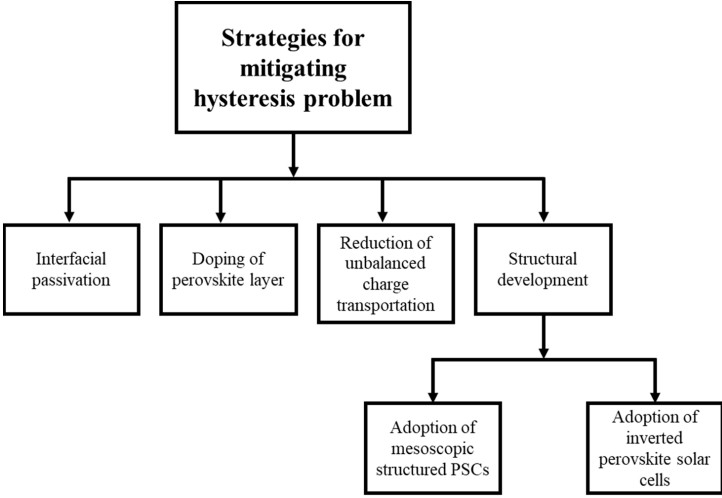

**Figure 7.** A summary of the strategies for mitigating or reducing hysteresis in PSCs.

3.3.1. Interfacial Passivation

Defects in the perovskite layer of a PSC device originate from interlayer ion migration when exposed to continuous light illumination or thermal stress under operating conditions [297]. The entire mechanism of ion migration and its effects on device stability have been discussed elsewhere (Section 2.1.2). It is believed that ion migration from the perovskite sensitizer to its surrounding layers is one of the most critical parameters for influencing the anomalous hysteresis of PSC devices [35,298,299]. Therefore, the passivation of the interfaces between the perovskite sensitizer and charge transport layers (CTLs) with an inert passivating material may become an effective strategy for reducing hysteresis from the devices. A few years ago, Zhao et al. [201] fabricated a mixed perovskite (i.e., $(FAPbI_3)_{0.85}(MAPbBr_3)_{0.15}$)-based device with an excellent PCE (18.05%) but considerable hysteresis. Introducing an extra modification layer of $Cs_4SnO_4$ almost eliminated the hysteresis of the device and enhanced its PV performance. Recently, Li et al. [15] introduced an ultrathin oxo-functionalized graphene/dodecylamine 2D nanosheet between the $Cs_{0.5}(FA_{0.85}MA_{0.15})_{0.95}Pb(I_{0.85}Br_{0.15})_3$ perovskite layer and the HTL (i.e., Spiro-OMeTAD) of their fabricated device. The intercalation of such an ultrathin passivation

layer enhanced the device stability by suppressing ion migration with an excellent PV performance (i.e., maximum PCE 21.1%, FF 81%) and lowered hysteresis under operating conditions. Another research group of Zhong et al. [296] reported the adverse effects of PCBM passivation layers on the ion migration of organic–inorganic hybrid lead halide PSCs (i.e., FTO/compact-$TiO_2$/$MAPbI_{3-x}Cl_x$/spiro-OMeTAD/Au, where x = 1,2,3). As an underlying mechanism, the authors ascribed the viability of the PCBM molecules to diffuse into the respective perovskite layer and passivate the defects that were previously created by $I^-$ ion migration. Thereby, a significant reduction in iodine ion/vacancies is obtained by introducing such an interlayer which suppresses further ion migration from the perovskite sensitizer. In another research work, Liu and his colleagues [182] reported the effect of introducing a halogen-containing multifunctional passivation layer (i.e., 4-chlorobenzamine and 4-fluorobenzamine) in their Cs-formamidium perovskite-based device. The electronegative halogen groups present in the passivating molecules significantly suppressed ion migration from the perovskite layer and reduced the regeneration of defects by the formation of strong ionic bonds with $Pb^{2+}$ ions and hydrogen bonds with the organic cations of the sensitizer as well.

### 3.3.2. Doping of Perovskite Layer

The misalignment of energy bands of the device layers is considered another significant factor behind the hysteresis of PSCs. This problem can be ameliorated by adjusting the energy bands of different layers by using selective contact materials or by doping perovskite and other charge transport materials with various dopants. Basically, three types of doping have been introduced in the literature so far, targeting the elimination of hysteresis from perovskite devices, e.g., (i) self-doping, (ii) ion-doping, and (iii) doping with guest molecules of the perovskite sensitizer. Self-doping is the strategy for the components forming a perovskite sensitizer to tune their energy-level position by enhancing or diminishing their relative percentage in the bulk perovskite compound. For instance, $MAPbI_3$, the most used perovskite material in PSC devices, is mainly composed of methylammonium halide (MAI) and lead iodide ($PbI_2$). The energy band of the respective perovskite film can be modulated by tuning the ratio between these precursor components. In brief, as the percentage of MAI increases compared to the $PbI_2$ components, the bulk perovskite turns into p-type material due to the facile formation of $Pb^{2+}$ ions. On the contrary, in $PbI_2$-rich perovskite sensitizers, the facile formation of $I^-$ vacancies makes it n-type material in nature [300]. A few years ago, Xie et al. [301] showed that the electronic structure of the $MAPbI_3$ absorber is directly correlated with the $MAI/PbI_2$ ratio in it and can be adjusted by tuning this ratio while fabricating the layer. Paul et al. [300] reported the gradual change in the electronic properties of both $MAPbI_3$ and $FAPBI_3$ perovskite sensitizers (i.e., shifting from p-type to n-type material) while using $PbI_2$-deficient precursors to $PbI_2$-rich precursors to fabricate the respective perovskite layers. In another research work, Wang et al. [302] showed that the $MAPbI_3$ absorber could be converted from p-type to n-type material by thermal annealing as the MAI parts of the material evaporate at a specific temperature.

In the ion-doping strategy, some selective monovalent and trivalent cations are used as dopants for amending the misalignments of the layer energy bands. As monovalent cationic dopants, the effectivity of alkali metal ions (i.e., $Na^+$, $K^+$, $Li^+$) and $Cu^+$ and $Ag^+$ ions has been introduced several times in the literature. Among them, doping $Cu^+$, $Ag^+$ and $Na^+$ ions in the $MAPbI_3$ perovskite layer by the Abdi-Jalebi group [303], doping $Na^+$ ions by Chang et al. [304], $Na^+$-doping in $CsPbBr_3$ nanocrystal perovskites by Li et al. [305] and in $FA_xMA_{1-x}PbI_3$-based mixed ionic perovskites by Okamoto et al. [306] demonstrated promising results in the suppression of hysteresis. Additionally, some trivalent cations such as $Sb^{3+}$, $In^{3+}$, etc., were also reported in the literature to obtain stabilized PV performance with an enhanced PCE [307]. Furthermore, molecular doping has undergone more experiments in recent years than the ion doping strategy for the purpose of band-bending of the perovskite sensitizers in PSC devices. The electronic properties of perovskite materials can be tuned to both n-type and p-type states by adding

different types of dopant molecules. Most recently, Jiang and his colleagues [308–310] introduced a small molecule of 4,′,4″,4‴-(pyrazine-2,3,5,6-tetryl) tetrakis (*N*,*N*-bios(4-methoxyphenyl)aniline) into the perovskite sensitizer ($FA_xMA_{1−x}PbI_3$) of their fabricated device and obtained a promising hysteresis-free PV performance (i.e., stabilized PCE ~23%) with it. Wu et al. [219] reported that a molecular dopant of 2,3,5,6-tetrafluoro-7,7,8,8-tetracyanoquinodimethane (F4TCNQ) can improve the conductivity of $MAPbI_3$ perovskite films by tuning its fermi level. Additionally, the favorable modification of $MAPbI_3$/ITO surface by the band-bending of the perovskite materials facilitated the carrier conductions through the interfaces which resulted in a highly stabilized PCE (i.e., above >20%), even without using any HTL. Chen et al. [303] reported the n-type doping of the $FACsPbI_xBr_{1−x}$ perovskite layer of their device by guanidinium bromide molecules which formed a graded perovskite homojunction on the top of the respective perovskite layer. The widened bandgap at the perovskite homojunction resulted in an improved $V_{oc}$ and a stabilized PV performance with a negligible hysteresis (Figure 8a). Recently, Zhang et al. [311] fabricated a guanidinium (GA)-doped multi-cationic perovskite-based device (i.e., $FTO/SnO_2/Cs_{0.05}(FA_{0.83}(MA_{1−x}GA_x)0.17)_{0.95}Pb(I_{0.83}Br_{0.17})_3(CsFAMA_{1−x}GA_x)/$ spiro-OMeTAD/Au) that exhibited over 20% PCE with a negligible hysteresis. Pham and his colleagues [310] reported the tunability of guanidinium iodides (GuI) for the hysteresis anomaly of a planar PSC device. As shown in Figure 8b–d), the device exhibited an enhanced PV performance (e.g., maximum PCE obtained 17.02%) with a tunable hysteresis on the basis of the amount of GuI dopants added to it.

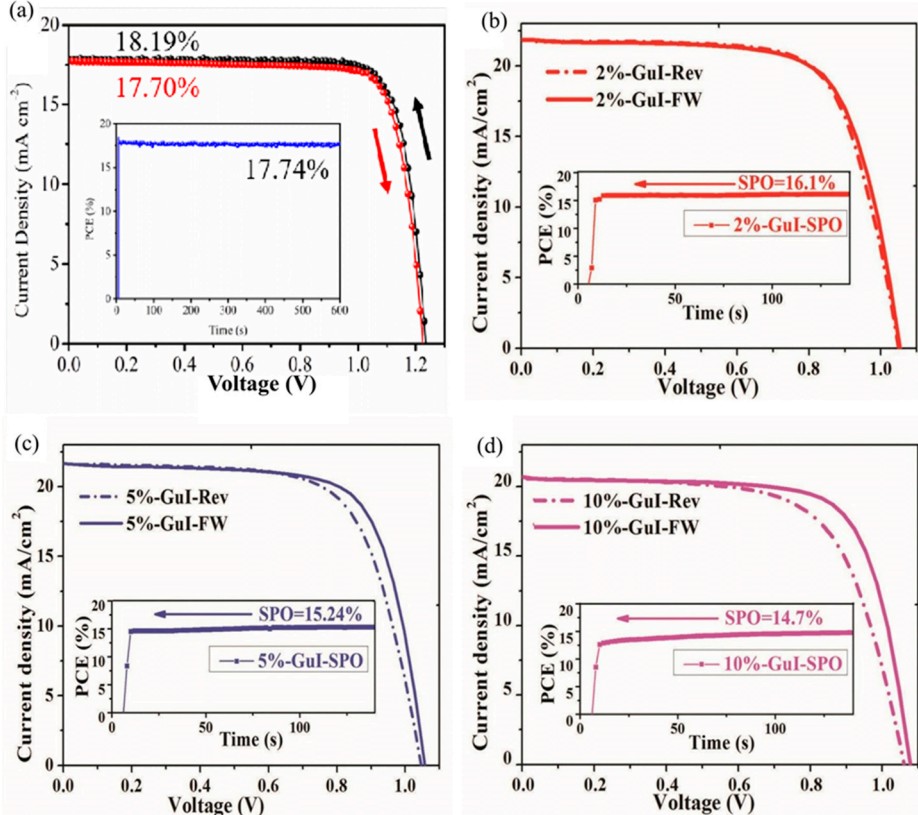

**Figure 8.** (**a**) J–V curves of the device with GABr treatment under different scans. The inset is the steady-state efficiency of the device (Reprinted with permission from Ref. [307]. Copyright: Elsevier 2019). The performance metrics of planar perovskite solar cells based on different compositions. Current density–voltage characteristics for the (**b**) 2%-GuI; (**c**) 5%-GuI; and (**d**) 10%-GuI-doped cells at scan rate of 150 mV s$^{-1}$ (Reprinted with permission from Ref. [312]. Copyright: WILEY-VCH Verlag GmbH & Co. KGaA, Weinheim 2019).

### 3.3.3. Reduction in Unbalanced Charged Transportation

Unbalanced charge transportation through the CTLs is considered another vital factor behind the hysteresis of PSC devices [44,45]. Usually, electrons transport at a faster rate than holes, despite being contemporaneously generated at the perovskite layer. Therefore, balancing these two opposite charges is a critical problem in the fabrication of PSC devices with selective materials. However, this problem can be resolved either by using alternative charge transport materials (CTMs) when the conventional materials give rise to hysteresis or by doping the CTMs with appropriate dopants. A few years ago, Chang et al. [311] propounded that a PCBM capping layer (as HTL) instead of the conventional spiro-OMeTAD HTLs can significantly suppress the leakage current that is produced under bias conditions and thereby, considerably reduce the device hysteresis by balancing the charge transportations through the CTLs. In another research work, Wang et al. [312] reported the doping of fullerene-based HTLs with pentace, i.e., a high-hole conductive material facilitates hole mobility through HTLs and helps balance the transportation of holes with electrons that considerably reduces the hysteresis. Recently, Shi et al. [213] demonstrated a supramolecular complex dopant (i.e., $[(C_8H_{17})_4N]_4[SiW_{12}O_{40}]$) for the $SnO_2$-ETL of their device and obtained almost a hysteresis-free device performance with long-time stability. Song et al. [313] noticed that the doping of conventional $SnO_2$-based nano-crystalline ETL with a rare earth element, namely yttrium (Y), significantly reduced the hysteresis of the device as well as considerably enhanced its PCE (up to the 20.71%). The excellent PV performance and comparatively lower hysteresis effect were attributed to the better charge transportation across the ETLs when doped with $Y^{3+}$-ions.

### 3.3.4. Structural Development

Apart from the electronic properties and energy level positioning of perovskites and the respective CTLs, the effect of the configuration and structure of the device on its hysteresis anomaly have also been reported by many researchers. For instance, mesoscopic structured perovskite devices have been proven to be less hysteric than the planar structured ones [212,314–316]. However, the drawbacks of depressed PV performance made them less promising for commercialization despite providing a long-term stabilized performance under operating conditions. Therefore, such devices would be more promising for commercialization if their PCE could have somehow been improved. In addition, the inverted planar structure of PSC devices can also considerably reduce the hysteresis of a PSC device compared to that of typical n–i–p structured devices [317,318]. However, selective CTMs should also be assured in that case as it is quite impossible to obtain a hysteresis-less device if energy bands of the device layers (i.e., HTL, ETL, perovskite absorber) do not match or become appropriately compatible with each other. Promising results were obtained with this strategy by Forgács et al. [319], Pandey et al. [46], Liang et al. [320], Hu et al. [279] and many other researchers. Such hysteresis-free behavior of the inverted planar devices was described by their electronically favorable charge extracting perovskite interfaces and consequently their less capacitive current freezing tendency compared to the standard n–i–p structured devices [317,319,320]. However, one major drawback of the inverted p–i–n structured devices is their lower PV performance compared to the standard ones [106,321–323]. Therefore, enhancing the PCE of inverted planar structured PSC devices will make them more promising in the future as they amicably reduce device hysteresis only by their identical configurations, without demanding any kinds of excessive costs or complexities for device engineering.

### 3.4. Sealing/Encapsulation

Air sensitivity along with the moisture sensitivity of perovskite materials emphasize the requirement of encapsulation to achieve a stable outdoor lifetime of over 25 years [40,157,324]. Even approximately 80% degradation of highly efficient perovskite PVs was reported in the literature when exposed to the environment of ambient conditions without encapsulation for highly efficient devices [325]. Generally, a thin glass coverslip is used to overcome such

undesired degradations, which is further sealed using UV curable epoxy resins [326–328]. Single-step high-temperature encapsulation (>150 °C) was reported to result in the delamination of active and metallization layers in devices. In contrast, the two-step encapsulation approach minimizes the defects that occur due to mechanical and thermal stress [329]. In a two-step encapsulation approach, the first step consists of making electrical contact and later embedding it in a polymer matrix to ensure mechanical stability. The next step involves laminating the pre-encapsulated device using low temperature, to retain the good PV characteristics of PSC. Sample devices are electrically monitored (*J–V* curves) during and after the process. Two types of configuration method are reported for encapsulation [330]. In one of them, the area between silver contact and plain glass cover is filled with UV curable epoxy resin while the other configuration comprises an electrode with a top glass cover and a gap between these two layers that contains a water-absorbent material for moisture absorption from the surrounding environments [331–334]. Many researchers also demonstrated the encapsulation of 2D PSC devices with plastic encapsulating barriers. This is typically available in two formats, namely partial and complete. However, such plastic encapsulation could only prevent moisture penetration and may stabilize the performance of PSC for more than one year [331,332,335,336]. Krebs et al. [337] developed an encapsulation method that was performed by attaching a 25 μm-thick PET barrier foil with an acrylic adhesive, thus improving device stability. Hwang et al. [338] improved the performance life of a device by spinning a layer of amorphous Teflon on top of the device. Indigorus et al. [339] introduced a conformal plasma polymer thin film by a simple solvent-free polymer encapsulation method for PSCs. Likewise, dozens of such materials have been reported as encapsulants for PSC devices including Teflon, $Al_2O_3$, poly (methyl methacrylate) (PMMA), and polycarbonate [13].

Ethylene-vinyl acetate is the most widely used encapsulating material among the single-layer encapsulation films that offers weather resistance and long-term reliability under extended periods of exposure to different degeneration-inducing components [340]. Furthermore, ethylene-methyl acrylate is another commonly used encapsulant with some excess advantages including its enhanced thermal stability, adherence to various substrates, chemical resistance, and good mechanical behavior at low temperatures [341]. Polyvinyl butyral was also reported to strongly reduce the moisture permeation into the device structure and ensure a longer device lifetime. Polymer cyclized perfluoropolymer, widely used in organic light-emitting diodes (OLEDs), could also be a good candidate with commercial interests as it provides a sophisticated barrier for commercially viable, flexible, and printable solar cells [342]. Additionally, the depositing of multi-layered encapsulants such as the Ta–Si–N/Ta–Si–O layer was also reported to significantly reduce the degradation of CIS solar cells at an accelerated aging test [343]. Some of the encapsulated devices, along with their enhanced stability, is summarized in Table 1.

**Table 1.** Encapsulated devices along with their stability.

| Device Structure | Testing Condition | Initial PCE | Remaining PCE | Ref. |
|---|---|---|---|---|
| FTO/*c*-TiO$_2$/*mp*-Al$_2$O$_3$/MAPbI$_3$xClx/spiroMeOTAD/Au | 1000 h at 40 °C | 12.3 | 50% | [94] |
| FTO/C60/mp-Al$_2$O$_3$/MAPbI$_{3-x}$Cl$_x$/spiroMeOTAD/Au | 500 h at 60 °C | 10.4 | 55% | [94] |
| FTO/PEDOT:PSS/(BA)$_2$(MA)$_3$Pb$_4$I$_{13}$/PCBM/Al | Under 1 sun 2250 h, 65% RH | 12.52 | 100% | [94] |
| FTO/*c*-TiO$_2$/*mp*-TiO$_2$/(FAPbI$_3$)$_x$ (MAPbBr$_3$)$_{1-x}$/spiroMeOTAD/Au | 3 months outdoors | 18.7 | 95% | [94] |
| FTO/LiNiO/Cs$_{0.05}$ FA$_{0.7}$MA$_{0.25}$PbI$_3$/C60/Al | 1 sun | 20.5 | 85% | [94] |
| FA$_{0.83}$Cs$_{0.17}$Pb(I$_{0.6}$Br$_{0.4}$)$_3$ | 4000 h in air | 17.5 | 80% | [94] |
| FTO/c-TiO$_2$/MAPbI$_3$/spiro-MeOTAD/Au | 500 h at 45 °C | 13.9 | 90% | [94] |
| FTO/TiO$_2$ (ZnO)/MAPbI$_3$/spiro-MeOTAD/MoO$_3$/Al | Under 1 sun 144 h, 85% RH | 14.0 | 85% | [344] |
| ITO/PEDOT:PSS/MAPbI$_{3-x}$Cl$_x$/PCBM/Ca/Ag | 2 month in ambient conditions | 12.25 | 90% | [345] |

## 4. Future Recommendations

Researchers have been working harder to improve the lifetime of PSCs. Notably, in the presence of UV-ray, moisture, $O_2$, and thermal stress, most of the suggested device configurations have failed to maintain long-term steady-state performance. However, some strategies have shown overwhelming outcomes in recent years. The priority in the future should be focusing on the following strategies:

- More intense research is needed to understand the degradation mechanisms and varying conditions for all types of perovskites such as $CH_3NH_3PbCl_3$, $CH_3NH_3SnI_3$, and $CH_3NH_3PbBr_3$.
- Research is required on the degradation mechanisms for all types of HTM layer and ETM layer under high thermal conditions.
- Investigation is needed to find the proper alternative to toxic Pb in perovskite to reduce environmental pollution. The alternative metal must be non-toxic and enhance the stability of PSC with high efficiency.
- Various organic modifiers must be applied to passivate the direct contact between the metal oxide films (ETLs) and the respective perovskite sensitizer to reduce the undesirable photocatalytic phenomena of the perovskite layer.
- Various potential bi-layer interfacial structures must be used instead of the single metal oxide-based ETLs to significantly reduce the metal ion diffusion and the charge recombination at the perovskite/ETL interfaces.
- Mixed-metal-based ETMs (such as $Zn_2SnO_4$ and La-doped $BaSnO_3$) must be developed with a focus on improving PCE in such devices.
- Developing alternative polymeric or inorganic p-type (i.e., hole-conductive) materials instead of the conventional spiro-OMeTAD that exhibits well-matched energy level alignment with the conventional perovskite materials, good hole mobility, conductivity, as well as better stability against moisture, oxygen, and thermal stress.
- Developing various non-hygroscopic dopants or additives that may help diminish the decomposition rate of conventional HTLs (i.e., spiro-OMeTAD).
- Enhancing PCE of HTL-free devices. For this purpose, gradient doping in carbon-based PSCs can be an effective solution for the future. More research should be provided in such devices to improve their PCE balancing and enhance stability.
- Focusing on the newer molecular doping strategies in the perovskite layer of PSCs to reduce the device hysteresis.
- Enhancing the PCE of inverted planar structured PSCs will make them more promising in the future as they amicably reduce device hysteresis only by their identical configurations.
- Focusing on the improvement of PCE in mesoscopic structured PSCs—such devices will be more promising than the planar structured devices in the future for commercialization if their PCE can be somehow improved.

## 5. Conclusions

The instability of the device is one of the main barriers to the commercialization of PSCs in the market. To overcome this issue, we reviewed the instability of PSCs under UV, light, air, humidity, thermal, and electric-field conditions in detail from both theoretical and experimental points of view. Studies on the stability of perovskite solar cells under different conditions have attracted growing concern. However, further research is nonetheless required. In this review, we introduced various degradation mechanisms, which produce fundamental knowledge and some awareness for stability enhancement. It was identified that, to enhance the stability of PSCs, several factors must be taken into account for their systematic engineering, including material engineering, novel device structure design, HTM, ETM layer, and electrode materials' preparation and encapsulation method. Among them, the partial substitution of the $MA^+$ cation with hydrophobic $BA^+$, $PEA^+$, $FA^+$, $Cs^+$ or mixed cation perovskite and the partial substitution of $I^-$ anion with $Cl^-$, $Br^-$, $SCN^-$, or multiple halogen perovskite can offer the enhanced stability under higher humidity and thermal conditions. The design of 2D perovskite can prevent the

direct contact of water with perovskite, which would chemically enhance the stability of perovskites in the air ambient. The suitable encapsulation also plays a significant role in improving the stability of perovskite devices to overcome the barrier to commercialization. Moreover, substituting the conventional Spiro-MeOTAD HTLs with various inorganic hole transport layers (CuI, CuSCN, NiO, and CuGaO$_2$) and even hole transport layer-free structures would significantly improve the stability of PSCs. SnO$_2$ can be the most promising candidate as an alternative to the conventional TiO$_2$-based ETLs to considerably enhance the device stability. However, enhanced intrinsic stability would stimulate the destination of commercialization, and proper encapsulation would ensure that devices fulfilled the desired lifetimes. Therefore, to obtain stable PSCs with a high PCE, simply modifying the present perovskite materials or interface is not adequate; we suggest establishing some new materials and designs with high stability under severe conditions. Solutions to enhance the stability and performance of PSCs have been described in this review; thus, strongly implementing those recommendations would lead to the required stability of the PSCs and can promote their commercialization.

**Author Contributions:** Formal Analysis, Writing—Original Draft Preparation: M.S.H. and J.A.; Writing—Original Draft Preparation: M.A.; Writing—Review & Editing: M.B.A. and A.S.; Writing—Review & Editing, validation: M.D.H. and J.M.; Validation, Supervision, Writing—Review & Editing: J.T.; Validation, Supervision, Writing—Review & Editing: M.A.H. All authors have read and agreed to the published version of the manuscript.

**Funding:** This research received no external funding.

**Institutional Review Board Statement:** Not applicable.

**Informed Consent Statement:** Not applicable.

**Data Availability Statement:** All data generated or analyzed during this study are included in this manuscript.

**Conflicts of Interest:** The authors declare no conflict of interest.

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
