# Peer review of "Recent Criterion on Stability Enhancement of Perovskite Solar Cells"

_processes, doi:10.3390/pr10071408_

Round 1

Reviewer 1 Report

Review Report. Technological Advancements in Nanomaterials Synthesis and Application:

This is a well presented report which is essentially a review of the academic and scientific literature on a topic ie options for better accessing solar power which has particular environmental and social interest. The discussion is well considered as are the conclusions regarding the next stages of the work.

While the report is well written, I suggest that immediately following the Introduction, the authors should add sections that i) articulate the Research Question and any hypotheses; and ii) articulate the proposed Methodology. Why it is suitable for this report, how the research is to be undertaken and the expectations for contributions to academic knowledge. These additions would then be maintaining the academic expectations for any research project.

Well done.

Author Response

Reviewer # 1

This is a well presented report which is essentially a review of the academic and scientific literature on a topic ie options for better accessing solar power which has particular environmental and social interest. The discussion is well considered as are the conclusions regarding the next stages of the work.

While the report is well written, I suggest that immediately following the Introduction, the authors should add sections that i) articulate the Research Question and any hypotheses; and ii) articulate the proposed Methodology. Why it is suitable for this report, how the research is to be undertaken and the expectations for contributions to academic knowledge. These additions would then be maintaining the academic expectations for any research project.

Well done.

General Response: We thank Reviewer # 1 for his/her broad review of the manuscript and appreciate the thoughtful comments that have been provided. We have now addressed the novelty of our work in the introduction section. We have articulated the expectations from this work for contributions to the academic knowledge as follows:

This report will help the researchers and academic personnel to get a broad view of the stability issues of PSCs and all the effective strategies to overcome these issues in brief. Besides, recommendations for further research are also provided that will help the researchers to improve this field more in future.”

Reviewer 2 Report

I found your paper to be a very interesting and comprehensive study. I only recommend you improve your document, which is very well written, correcting only a few sentences, avoiding repeated words and completing or correcting their meaning.

Author Response

Reviewer # 2

I found your paper to be a very interesting and comprehensive study. I only recommend you improve your document, which is very well written, correcting only a few sentences, avoiding repeated words and completing or correcting their meaning.

General Response: We thank Reviewer 2 for his/her review of the manuscript and appreciate the thoughtful comments that have been provided. We have thoroughly revised our manuscript according to the suggestions, corrected some sentences and excluded repeated words.

Reviewer 3 Report

Comments for authors

 The stability issues of perovskite solar cells is a major problem in its upscaling. Many reviews related to degradation pathways in PSCs and its improvement has been published. I have some queries which as follows:

1.  Herein, authors have discussed exposure of external factors (e.g. heat, moisture, and UV light) that creates PVs instability, and their mechanisms to induce such poor device stability. Thus, abstract should include the objective of this review.

2. A recent review on Recent trends in efficiency-stability improvement in perovskite solar cells (DOI: 10.1016/j.mtener.2020.100449)has also discussed the role of ETLs and HTLs. What is different in this manuscript?

3. Latest research of 2022 should also be added.

Author Response

Reviewer # 3

The stability issues of perovskite solar cells is a major problem in its upscaling. Many reviews related to degradation pathways in PSCs and its improvement has been published. I have some queries which as follows:

  1. Herein, authors have discussed exposure of external factors (e.g. heat, moisture, and UV light) that creates PVs instability, and their mechanisms to induce such poor device stability. Thus, abstract should include the objective of this review.
  2. A recent review on Recent trends in efficiency-stability improvement in perovskite solar cells (DOI: 10.1016/j.mtener.2020.100449)has also discussed the role of ETLs and HTLs. What is different in this manuscript?
  3. Latest research of 2022 should also be added.

General Response: We thank Reviewer 3 for his/her review of the manuscript and appreciate the thoughtful comments that have been provided. We have addressed the issues as below:

Comment 1: Herein, authors have discussed exposure of external factors (e.g. heat, moisture, and UV light) that creates PVs instability, and their mechanisms to induce such poor device stability. Thus, abstract should include the objective of this review.

Response 1:

We thank the reviewer for his/her suggestion. We have edited our abstract according to this comment and included the objective of this review as follows:

“The mechanism for device degradation for several parameters and the complementary materials showing promising results are systematically analyzed. The main objective of this work is to review the effectual strategies of enhancing the stability of PSCs. Several important factors such as material engineering, novel device structure design, hole transporting materials (HTMs), electron transporting materials (ETMs), electrode materials preparation, and encapsulation methods that need to be taken care of in order to improve the stability of PSCs are discussed extensively.”

Comment 2: A recent review on Recent trends in efficiency-stability improvement in perovskite solar cells (DOI: 10.1016/j.mtener.2020.100449) has also discussed the role of ETLs and HTLs. What is different in this manuscript?

Response 2: We thank the reviewer for this comment. The mentioned article “Recent trends in efficiency-stability improvement in perovskite solar cells” focused on different structures of perovskite solar cells (PSCs) with different ETLs and HTLs; and just reviewed the obtained PCEs and performance stability of those structures.

In our article, we have focused on different strategies associated with different layers of PSCs to overcome their stability issues, which is, however, not discussed in the above-mentioned review article. For instance, in our article, we have discussed about the comparative stability of PSCs using (i) single layered ETMs, (ii) bi-layered ETMs, (iii) various doped ETLs; as well as the effect of (iv) interfacial passivation between ETLs and perovskite layers, and (v) the structural alteration of ETLs on the stability enhancement of PSCs. In the discussion about HTLs, we have reviewed the effects of using (i) non-hygroscopic additives and (ii) different alternative HTLs instead of the most conventional Spiro-OMeTAD and PTAA-based HTLs; on the stability enhancement of PSCs. Besides, we have extensively discussed about different stability problems of PSCs, e.g., (i) chemical instabilities (instability due to UV exposure, moisture, oxygen; ion migration problems) and (ii) thermal instabilities (thermal degeneracy of perovskite crystal structure, HTLs and ETLs). These issues are not discussed so broadly in the mentioned article. In addition, the mentioned article did not discuss about the hysteresis problem of PSCs which is one of their most significant undesired labyrinths. We have broadly discussed the sources, reasons and different strategies of mitigating this issue also in our article.

Comment 3: Latest research of 2022 should also be added.

Response 3: We thank the reviewer for this suggestion. We have now added some recent reference works of 2022 (reference no. 23, 34, 36, 39, 40, 43, 44, 45, 46, 147, 184, 194)

Round 2

Reviewer 3 Report

Authors have efficiently up-dated the review and it clearly describes the latest research related to parameters of PSC’s device instability.